# ST-D3DDARN: Urban traffic flow prediction based on spatio-temporal decoupled 3D DenseNet with attention ResNet

**Jing Chen**[☯], **Guowei Yang**[iD]*[☯], **Zhaochong Zhang, Wei Wang**

College of Information Technology and Engineering, Tianjin University of Technology and Education, Tianjin, China

☯ These authors contributed equally to this work.
* ygw@tute.edu.cn

**Data Availability Statement:** All code and dataset files are available from the github database (accession number(s) https://github.com/761049669/ST-DDDARN)

## Abstract

Urban traffic flow prediction plays a crucial role in intelligent transportation systems (ITS), which can enhance traffic efficiency and ensure public safety. However, predicting urban traffic flow faces numerous challenges, such as intricate temporal dependencies, spatial correlations, and the influence of external factors. Existing research methods cannot fully capture the complex spatio-temporal dependence of traffic flow. Inspired by video analysis in computer vision, we represent traffic flow as traffic frames and propose an end-to-end urban traffic flow prediction model named Spatio-temporal Decoupled 3D DenseNet with Attention ResNet (ST-D3DDARN). Specifically, this model extracts multi-source traffic flow features through closeness, period, trend, and external factor branches. Subsequently, it dynamically establishes global spatio-temporal correlations by integrating spatial self-attention and coordinate attention in a residual network, accurately predicting the inflow and outflow of traffic throughout the city. In order to evaluate the effectiveness of the ST-D3DDARN model, experiments are carried out on two publicly available real-world datasets. The results indicate that ST-D3DDARN outperforms existing models in terms of single-step prediction, multi-step prediction, and efficiency.

## Introduction

With the progress of urbanization, the urban population and traffic flow are increasing rapidly. Accurate and efficient prediction of urban traffic flow holds significant importance in areas such as traffic management, public safety, and travel planning [1–3]. For instance, according to data released by the Chinese Ministry of Transport, the losses caused by traffic congestion account for 20% of the per capita disposable income in urban areas. On October 30, 2022, a Halloween party held in Itaewon, Seoul, South Korea, resulted in a massive stampede, leading to 159 casualties. This incident marked the ninth-largest stampede of the 21st century. Major cities like Beijing and New York face daily traffic congestion, causing severe economic losses, environmental pollution, and public safety issues. If the traffic management department obtains the results of traffic flow prediction [4] in advance and guides the traffic flow and

**Funding:** The Foundation for development of Science and Technology of Tianjin Education Committee under Grant No.2021KJ008 and W W, C J. 20220105 Tianjin Jinnan District Bureau of Science and Technology.

**Competing interests:** The authors have declared that no competing interests exist.

crowd in time, it can reduce the occurrence of congestion, stampede and other events to a certain extent. Therefore, achieving accurate and efficient predictions of the city's traffic flow holds crucial practical significance.

This study aims to predict future urban traffic flow through the analysis of extensive offline GPS data, including data from bicycles, taxis, and other sources. However, these GPS data possess temporal and spatial attributes, and efficiently and comprehensively mining the spatio-temporal correlations within them poses a significant challenge in establishing high-performance traffic flow prediction models. In recent years, numerous researchers have employed data-driven methods [5] for traffic prediction modeling, which can be broadly categorized into two types: traditional machine learning methods and deep learning methods. Traditional machine learning methods demand high requirements for feature engineering and struggle to handle high-dimensional traffic flow data, leading to limited overall applicability.

Fortunately, with the development of deep learning, modeling high-dimensional traffic data has become achievable, enabling the capture of complex features through a hierarchical approach. The earliest deep learning method used for traffic flow prediction is Recurrent Neural Network (RNN) [6]. However, when dealing with data featuring long-term dependencies, RNN encounters the issues of gradient explosion or vanishing gradients [7]. Consequently, researchers proposed its variants: Long Short-Term Memory (LSTM) [8] and Gated Recurrent Unit (GRU) [9]. Nevertheless, they require continuous time series as input and, when dealing with spatial data, necessitate data dimensionality reduction, thereby overlooking the spatial correlations in the data. Convolutional Neural Networks (CNN) can automatically and hierarchically capture the spatial features of traffic flow through convolution operations. Therefore, many researchers have built upon the two-dimensional convolutional neural networks(2D CNN) to propose deep composite networks [10–14] for extracting the spatio-temporal correlations of traffic flow. However, regional traffic flow may be influenced by the traffic flow in different regions at adjacent time points. Methods based on 2D CNN have certain limitations in establishing spatio-temporal correlations.

Due to the capability of 3D CNN in capturing spatio-temporal features, it has been widely applied in video analysis [15–17]. We posit that video analysis and the analysis of urban traffic flow changes share similar spatio-temporal correlations, as illustrated in **Fig 1**. Therefore, it is a feasible method to apply 3D CNN to urban traffic flow prediction. In this study, we process taxi GPS data and shared bicycle rental data into traffic frames, where each frame represents traffic flow at fixed time intervals (shown in **Fig 2**). However, in our experiments, we observed that 3D CNN faces challenges in extracting spatio-temporal correlations, including high computational complexity and difficulties in convergence.

Additionally, in comparison to video analysis, traffic flow prediction exhibits the following distinctive characteristics:

- **Multiple Temporal Correlations:** The traffic flow in a given period is not only related to the traffic flow in the nearby periods but also correlated with the traffic flow in the corresponding periods of previous days and weeks. (i.e., closeness, period, trend).;

- **Complex Spatial Correlations:** Traffic flow across all regions of the city mutually influences each other. With the advancement of urbanization, transportation becomes more convenient and people can quickly get to their destinations. Consequently, the traffic flow in one area is influenced by various regions within the city;

- **Heterogeneity:** Different regions around a location have different influences on it, and the influence of the same location on the surrounding regions is also different at different times. For example, road segments around school experience traffic congestion during students'

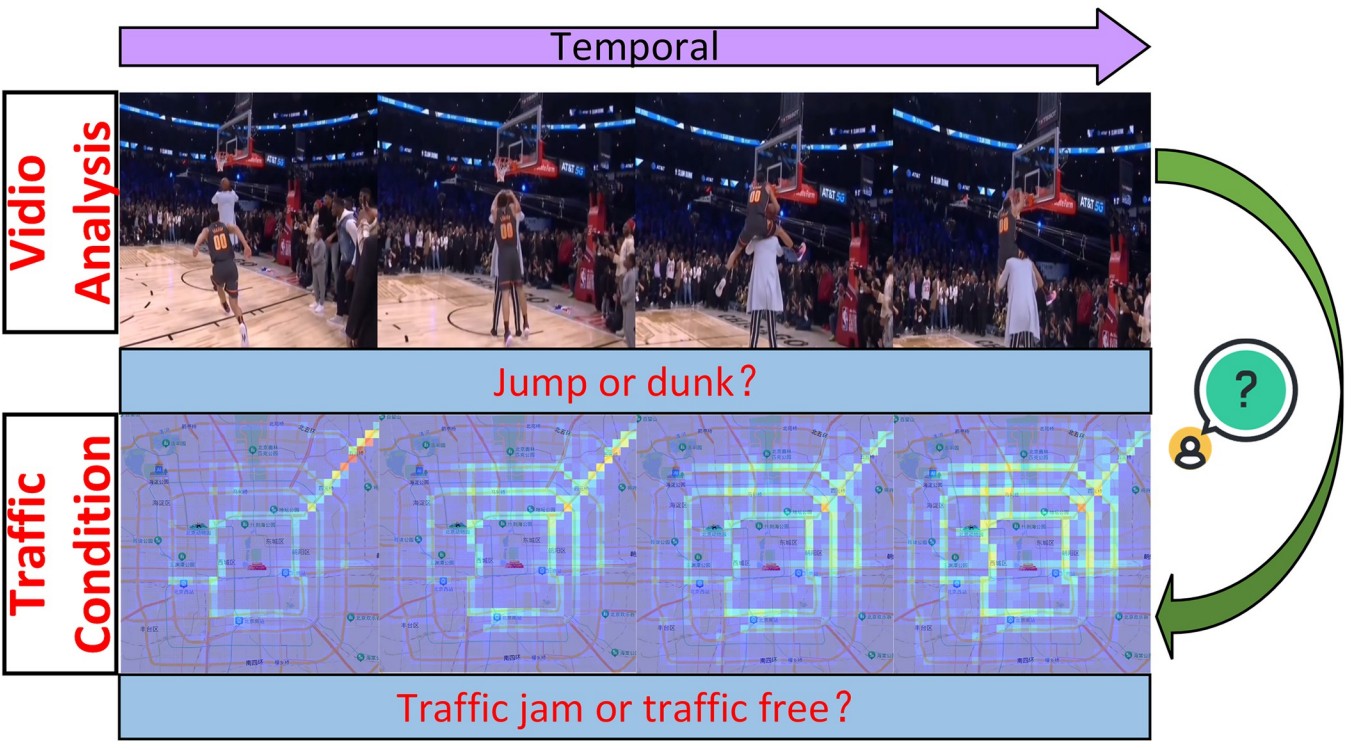

**Fig 1. Source of ideas in this paper.**

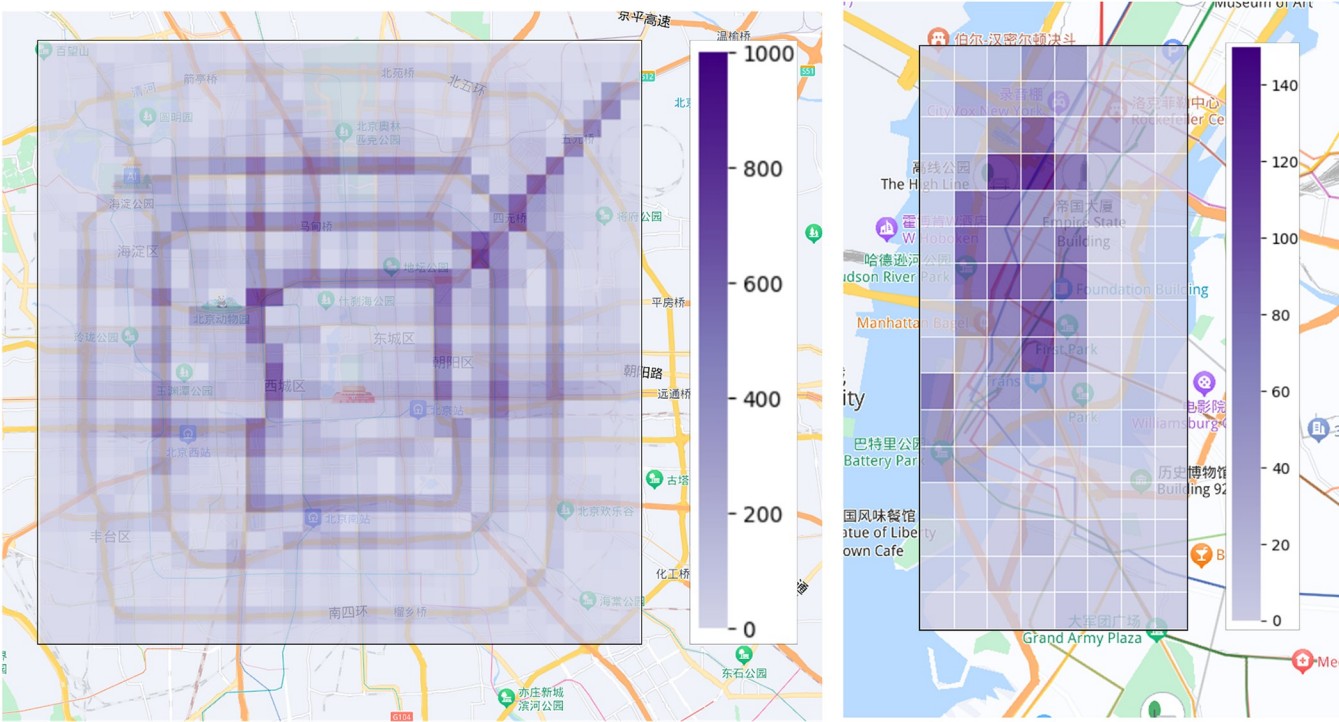

**Fig 2. Visualization of traffic flow maps for TaxiBJ, BikeNYC datasets.**

dismissal, while suburban areas are minimally affected by traffic flow from surrounding regions. Additionally, the importance of areas with tourist attractions is higher during holidays, diminishing during the tourism off-season;

- **External Factors:** Urban traffic flow is also influenced by various external factors, such as weather conditions, holidays, and special events.

- To address the challenges in urban traffic flow prediction mentioned above, the ST-D3DDARN has been proposed. The predictive model built upon this network exhibits high forecasting accuracy and low computational complexity. The innovations and primary contributions are as follows:

- Four branches, namely closeness, period, trend, and external factors, are established to extract multi-source information for traffic flow. Additionally, a unique keyframe construction is designed specifically for the period and trend branches;

- A dense decoupled 3D CNN is introduced. Experimental results indicate that the decoupled 3D CNN effectively addresses the challenges of long training times and slow convergence associated with 3D CNN. The dense connection network captures multi-level features, establishing a multi-scale spatio-temporal dependency. In essence, this network captures multi-scale spatio-temporal correlations with fewer parameters;

- A residual network incorporating both spatial self-attention and coordinate attention mechanisms is devised. The spatial self-attention mechanism dynamically captures global spatial correlations, while the coordinate attention quantifies the regional contributions between channels to address the heterogeneity of traffic flow;

- Through single-step prediction, direct multi-step prediction, recursive multi-step prediction, model efficiency evaluation, model prediction error visualization and other experiments, the superiority of the proposed method is verified.

In summary, the network design encompasses multiple innovative components, such as the unique keyframe construction, dense decoupled 3D CNN, and the integration of spatial and coordinate attention mechanisms in a residual network. Comprehensive experiments demonstrate the effectiveness and superiority of the proposed methods in various aspects of traffic flow prediction.

## Related work

### Traffic prediction

In recent years, traffic congestion has gradually become a prominent societal issue, prompting in-depth research into traffic flow prediction. This section reviews the research achievements in traffic prediction over the past few years.

Traditional methods for traffic flow prediction primarily relied on Historical Averages (HA), Support Vector Machines (SVM), AutoRegressive Integrated Moving Average (ARIMA) [18], and similar techniques. With the development of deep learning, Recurrent Neural Networks (RNN) [6] gradually replaced traditional methods. Some researchers explored variants of RNN, such as Long Short-Term Memory (LSTM) and Gated Recurrent Unit (GRU), for traffic flow prediction [19–22]. While these methods effectively capture temporal correlations, spatial correlations between regions have not been fully utilized.

In order to capture the complex spatial correlations in traffic flow, researchers employed different modeling strategies to simulate various spatial interactions. Zhang et al. [23] transformed historical trajectory data into image data, proposing a spatio-temporal data prediction

model (DeepST) that predicts the population flow throughout the city by overlaying convolutional layers. Subsequently, Zhang et al. [10] utilized multiple residual convolutional unit branches to model spatial properties from closeness, period, and trend aspects, demonstrating good prediction performance. Lin et al. [11] introduced DeepSTN+, employing 2D CNN and ResPlus modules to extract remote dependencies in traffic flow effectively. Wang et al. [12] aggregated features like closeness, period, and trend into multi-channel features input to a single-branch residual network, showing high simplicity and superior predictive performance. Ding et al. [13] designed a Deep Interlaced Training Network (MS-ResCnet) to improve traffic flow prediction performance further. To capture richer spatio-temporal correlations, Dai et al. [14] proposed multi-perspective convolution, convolving spatio-temporal features from the front, side, and top perspectives. The above methods all use 2D CNN to model traffic flow prediction, which has certain limitations in establishing spatio-temporal correlation.

In order to better simulate the spatio-temporal correlations of traffic flow, researchers began exploring methods that combine CNN and RNN. Yao et al. [24] introduced a Spatiotemporal Dynamic Network (STDN), combining LSTM, 2D CNN, and periodic attention transfer to dynamically simulate the spatio-temporal correlations of traffic flow. 3D CNN has a stronger ability to capture spatio-temporal features than 2D CNN and is widely used in video analysis. Guo et al. [25] used 3D convolution to simultaneously capture the temporal and spatial correlations of traffic flow. Chen et al. [26] proposed a Multi-Gated Spatio-temporal CNN (MGSTC), extracting various spatio-temporal features from traffic data through multiple gated 3D CNN branches. Zhou et al. [27] established a residual network using 3D CNN and constructed a filtering space attention block to dynamically adjust spatial weights. He et al. [28] introduced a Long Short-Term Spatio-temporal Feature Extraction Module, 3D-ConvLSTMNet. This method captures short-term spatio-temporal correlations through 3D CNN, utilizes ConvLSTM to capture long-term spatio-temporal correlations, and employs a residual structure to obtain long-distance spatial dependencies.

However, the above methods did not consider multi-scale spatio-temporal dependencies. Additionally, using 3D convolution to establish spatio-temporal correlations incurs high computational costs and slow convergence. Our proposed method effectively addresses these shortcomings and efficiently learns the complex spatio-temporal dependencies in traffic flow.

## Attention mechanism

The attention mechanism is an inherent ability in human vision that helps us quickly focus on key content, and it is now widely applied in the field of deep learning. SE-Net [29] captures inter-channel information through two-dimensional global pooling, significantly enhancing model performance with lower computational costs. However, SE-Net only focuses on inter-channel information, neglecting intra-channel spatial information. CBAM [30] integrates both channel and spatial information, demonstrating superior performance compared to SE-Net. Nevertheless, CBAM's attention to spatial information is based on the average and maximum values of each channel at each position, potentially resulting in the loss of inter-channel information. In the context of traffic frame-based traffic flow prediction tasks, it is essential to extract global dependencies in both time and space simultaneously. CA-Net [31], as an innovative and efficient attention mechanism, captures spatial information across channels, enabling the model to accurately identify the features of interest.

In the field of traffic prediction, attention mechanisms are employed to address the insufficient extraction of spatio-temporal features from traffic data. For example, Shi et al. [32] proposed a attention-based periodic time network, effectively capturing the spatial and periodic features of road networks through an encoder attention mechanism. Zheng et al. [33]

introduced a self-attention graph convolutional network that dynamically captures spatial correlations on a global scale. The Transformer [34] is a highly parallel self-attention mechanism that efficiently handles sequence data. Pu et al. [35] introduced a multi-view spatio-temporal Transformer (MVSTT) network, which dynamically captures spatial correlations and long-term dependencies in traffic flow from multiple perspectives.

In summary, attention mechanisms enhance the model's nonlinear fitting capability, effectively and comprehensively establishing spatio-temporal correlations in traffic data.

## Problem definition and analysis

In this section, we will introduce the core definition of urban area traffic flow prediction and analyze several key characteristics associated with it.

**Definition 1 (Grid Area $M_{i,j}$):** The city is divided into an i × j grid map based on latitude and longitude, where each grid represents a specific area of the city (see **Fig 3(A)**). $M_{i,j}$ denotes the grid located in the i-th row and j-th column of the grid map.

**Definition 2 (Traffic Time Slice $M_{i,j}^t$):** It records traffic data within a fixed time interval (1 hour or 30 minutes) and transforms it into image data $M_{i,j}^t \in \mathbb{R}^{P \times I \times J}$, where P, I, and J represent the channels, width, and height of the image, respectively. Since traffic flow consists of inflow and outflow (see **Fig 3(B)**), the number of channels is fixed at 2.

**Definition 3 (Inflow and Outflow $M_{in,i,j}^t, M_{out,i,j}^t$):** Inflow and outflow represent the number of units entering (leaving) a specific area within a fixed time interval. The inflow and outflow of the grid $M_{i,j}$ are defined as follows:

$$M_{in,i,j}^t = \sum_{Tr \in U} |\{\alpha > 1 | g_{\alpha-1} \notin M_{i,j} \wedge g_\alpha \in M_{i,j}\}| \tag{1}$$

$$M_{out,i,j}^t = \sum_{Tr \in U} |\{\alpha > 1 | g_\alpha \in M_{i,j} \wedge g_{\alpha+1} \notin M_{i,j}\}| \tag{2}$$

Where $Tr : g_1 \rightarrow g_2 \rightarrow g_3, \ldots, \rightarrow g_{|Tr|}$ represents trajectory data in U, U is the set of trajectory data in a fixed time interval, and $g_\alpha$ represents the latitude and longitude coordinates of GPS or sensor locations.

**Definition 4 (Urban Area Traffic Flow Prediction):** Urban area traffic flow prediction is divided into single-step prediction and multi-step prediction.

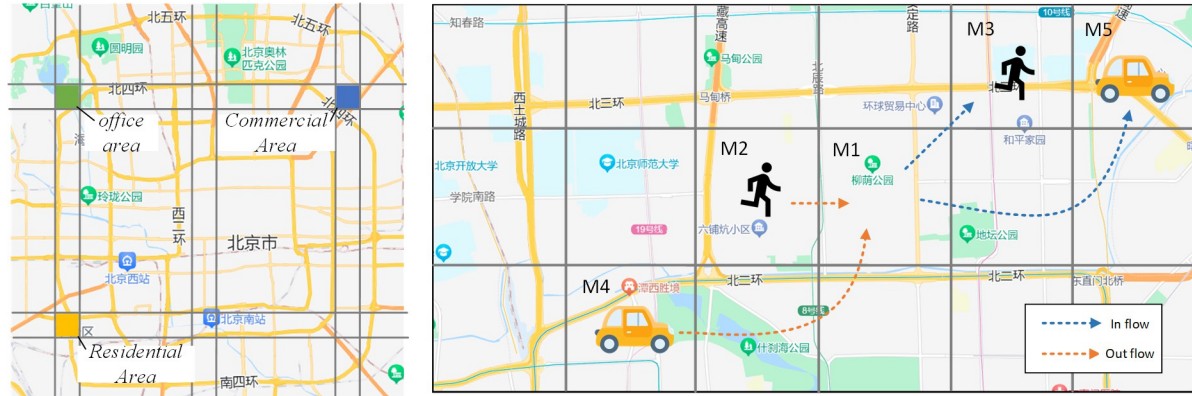

(a) Urban grid map division          (b) Diagram of inflow and outflow of region M1

**Fig 3. Urban gridding and definition of inflow and outflow.**

- **Single-step prediction:** Predict the inflow and outflow of each area in the next time period. In other words, $N = \{M^1, M^2, M^3, \cdots, M^t\}$ represents historical observed values, and $M^{t+1}$ represents the target prediction values.

- **Multi-step prediction:** Predict the inflow and outflow of each area in the n-th future periods, where $N = \{M^1, M^2, M^3, \cdots, M^t\}$ represents historical observed values, and $M^{t+n}(n>1)$ represents the target prediction value, and n is the number of prediction steps. Multi-step prediction is further divided into direct multi-step prediction and recursive multi-step prediction.

  1. **Direct multi-step prediction:** Obtains the target prediction values by training a new model. However, this approach has significant drawbacks, such as the need to train multiple models for different time steps, resulting in a waste of computational resources;

  2. **Recursive multi-step prediction:** Uses the prediction value of the previous time step as the input for predicting the next time step. This method only requires training a single model, saving computational resources through recursion.

**Analysis 1 (Complex Spatial Correlations):** The traffic flow in a specific area is influenced not only by nearby regions but also by regions at a considerable distance.

- **Local Spatial Correlation:** Traffic flow is impacted by the flow in nearby areas. As depicted in **Fig 3(B)**, the inflow traffic to M1 is influenced by the outflow traffic from M2 and M3. Simultaneously, the outflow traffic from M1 causes changes in the inflow traffic of M2 and M3.

- **Remote Spatial Correlation:** Traffic flow between remote regions also mutually influences each other. As shown in **Fig 3(B)**, residents commuting through highways or subway lines, such as M5 to M1 or M1 to M4, enable rapid interaction of long-distance traffic flow, leading to the traffic flow being influenced by regions at a considerable distance.

**Analysis 2 (Multiple Temporal Correlations):** The traffic flow at a specific moment is influenced by the traffic flow from past moments, exhibiting multiple dependency relationships, including closeness period and trend.

- **Closeness:** Predicting traffic flow is influenced by the traffic flow from the most recent moments. For example, the traffic flow at 8 PM is affected by the traffic congestion at 7 PM.

- **Period:** Traffic flow at the same time on consecutive workdays tends to be similar. For instance, there is a morning rush hour on every workday. As observed in, consecutive workdays exhibit distinct periodicity.

- **Trend:** Traffic flow exhibits a peak shift phenomenon every week. For instance, with the onset of winter, the earlier sunset time leads to an advance in the evening peak hours.

**Analysis 3 (External Factors):** External factors such as weather conditions, holidays, and special events have a significant impact on urban traffic flow. As depicted in **Fig 4**, there are different trends in traffic patterns between weekdays and weekends. Weekdays exhibit morning and evening rush hours, while traffic flow on weekends tends to be relatively smooth. Additionally, adverse weather conditions, leading to slippery roads and reduced visibility, contribute to lower traffic flow. **Fig 5** illustrates a traffic flow comparison between rainy and clear days in the Sanyuan Bridge area of Beijing.

**Processing External Factor Data:** External factor data comprises holiday and weather data. Holiday data includes various categories such as weekends, National Day, Labor Day, etc.

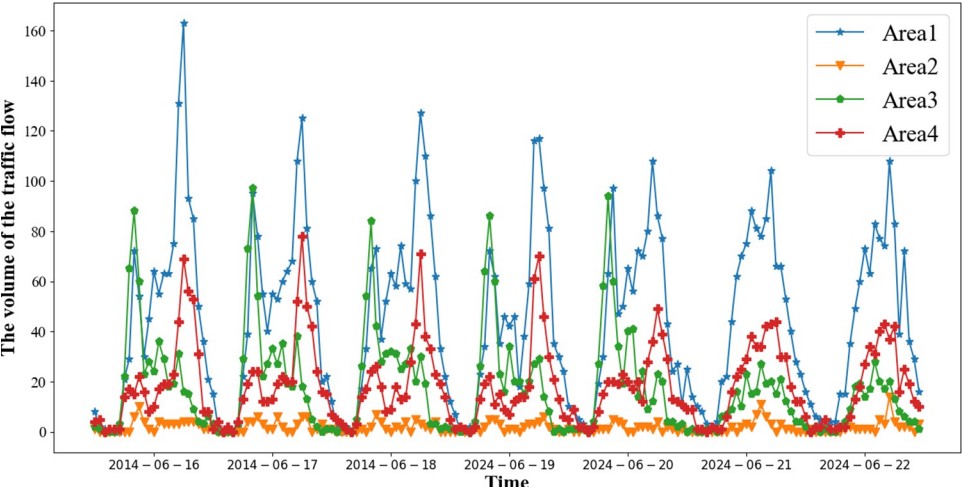

**Fig 4. Inflow flows in four zones over a week.**

Weather data encompasses conditions like weather status, temperature, wind speed, and atmospheric pressure. Categorical variables are digitized through one-hot encoding, while continuous variables are normalized within the [−1, 1] range.

## The proposed method

In this section, we introduce the various modules of the ST-D3DDARN model, and the overall structure of the model is illustrated in **Fig 6**.

The ST-D3DDARN model initially employs three branches (Closeness Branch, Period Branch, Trend Branch and External Factors Branch) to extract features related to closeness, period, trend and external factors of traffic flow, respectively. Subsequently, an Attention Residual Network is used to integrate these features and dynamically establish global spatial correlations.

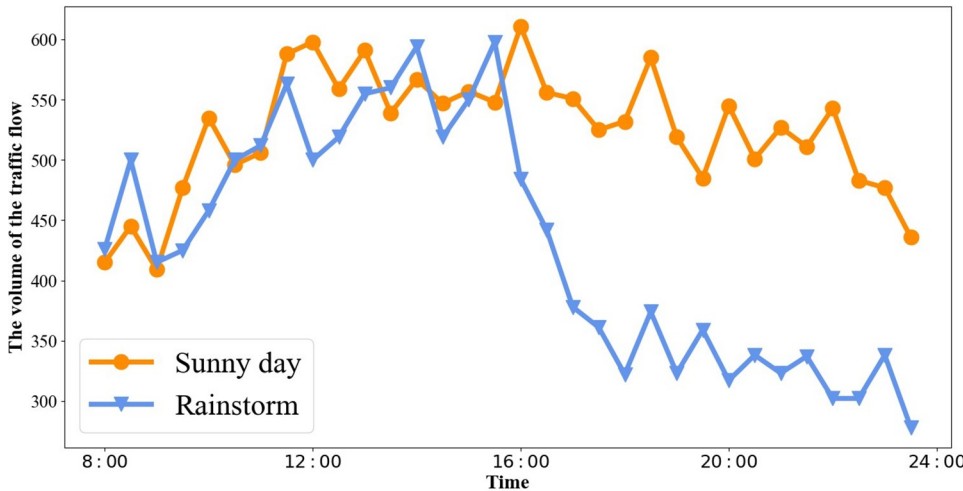

**Fig 5. Comparison of traffic flow on sunny day and rainstorm.**

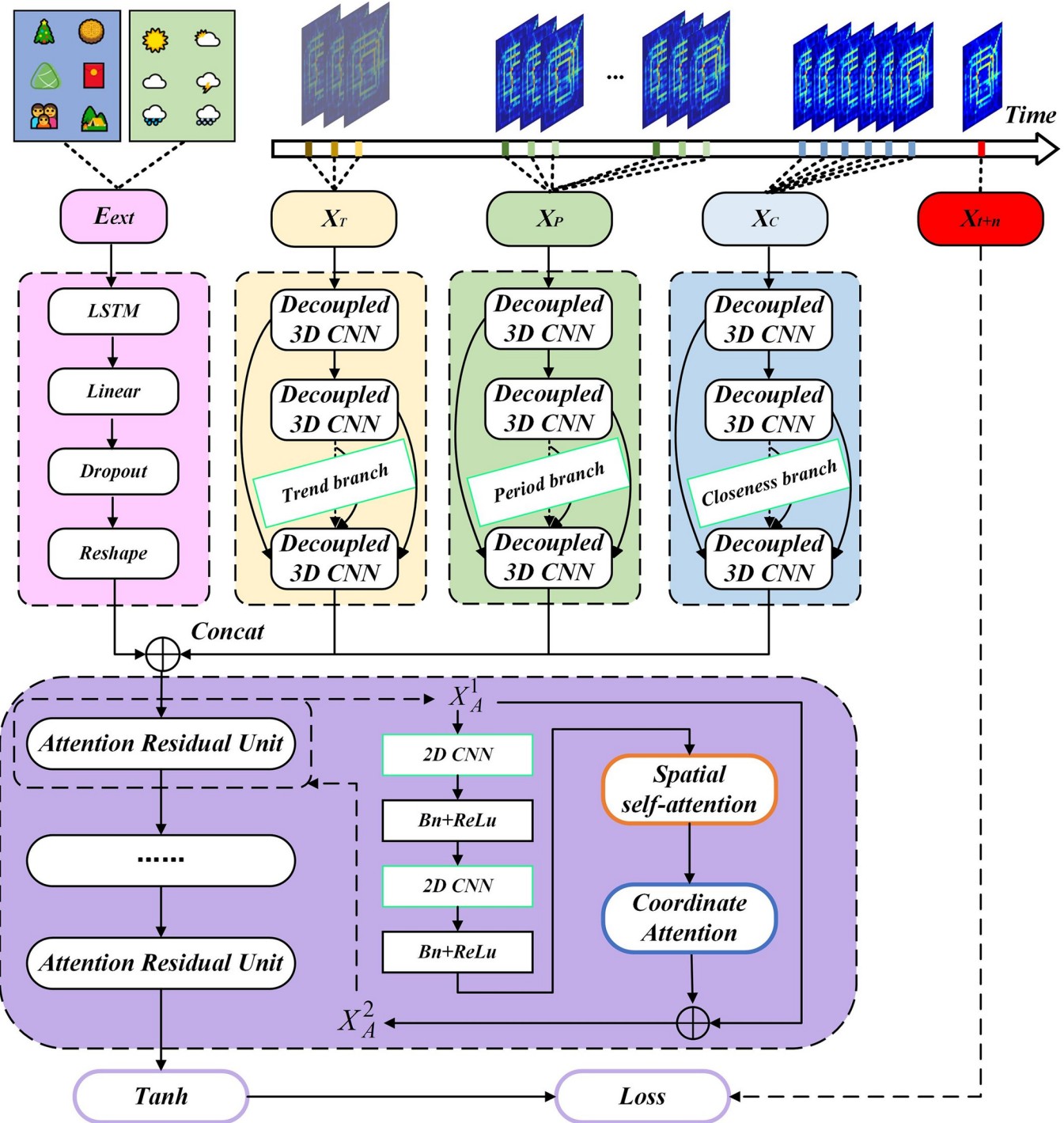

**Fig 6. The overall framework of the ST-D3DDARN model.**

The closeness branch period branch and trend branch extract multi-scale spatio-temporal correlation through densely connected decoupled 3D convolution, and the external factor branch extracts external information through LSTM and fully connected layers. The specific input features for the closeness, period, trend and external factors are as follows:

**Closeness:** The traffic frames for the preceding c time intervals of the target prediction period, represented as $X_C = [M^{t-c}, M^{t-(c-1)}, M^{t-(c-2)}, \cdots, M^{t-1}]$, where $c$ represents the length of the closeness sequence;

**Period:** Due to the translation phenomenon in traffic flow (which refers to the external events causing the trend of traffic flow to advance or lag), using only a single traffic frame makes it difficult to learn the periodicity of traffic flow. Therefore, we supplement the previous and subsequent frames, considering the three frames together as the traffic state at a specific moment. The input to the periodic branch can be expressed as $X_P = [M_P^{t-l_d \cdot d_{daily}}, M_P^{t-(l_d-1)d_{daily}} \cdots, M_P^{t-d_{daily}}]$, where $M_P^n = [M_P^{n-1}, M_P^n, M_P^{n+1}]$, $d_{daily}$ represents the total number of time slots within a day, and $l_d$ represents the length of the period sequence;

**Trend:** Similarly, the input to the trend branch can be expressed as $X_T = [M_T^{t-l_w \cdot w_{weekly}}, M_T^{t-(l_w-1)w_{weekly}} \cdots, M_T^{t-w_{weekly}}]$, where $M_T^n = [M_T^{n-1}, M_T^n, M_T^{n+1}]$, $w_{weekly}$ represents the total number of time slots within a week, and $l_w$ represents the length of the trend sequence;

**External Factors:** The external features of the first e periods of the target prediction period form a two-dimensional feature matrix $E_{ext} = [E_{ext}^{t-l_e}, E_{ext}^{t-(l_e-1)}, E_{ext}^{t-(l_e-2)}, \cdots, E_{ext}^{t-1}]$, where $l_e$ represents the length of external factors sequence.

The input to the attention residual network comprises features extracted from four branches. It dynamically establishes global spatial correlations through a spatial self-attention, quantifies the contribution of features between channels using CA-Net, and establishes residual connections. This network is effective in handling the heterogeneity of traffic flow.

## 3D convolutional neural network

Learning sufficient spatio-temporal information is crucial for traffic prediction. When using 2D CNN for convolution on traffic frames, the operation is limited to the spatial dimension, making it challenging to establish spatio-temporal correlations across consecutive frames. In contrast, 3D CNN has proven effective in extracting spatio-temporal features in video analysis [15, 16]. **Fig 7** illustrates the comparison between 3D CNN and 2D CNN. In **Fig 7(A)**, convolution on features can only establish connections in the spatial dimension. In **Fig 7(B)**, 3D CNN, equipped with three-dimensional filters, can capture spatio-temporal features across consecutive frames. Therefore, treating traffic flow as traffic frames and utilizing 3D CNN to capture spatio-temporal correlations proves to be an effective approach. The computation process of 3D convolution is outlined as follows:

$$m_{i,j}(x,y,z) = \sum_{l,m,n} M_i^{in}(x-l, y-m, z-n)W_j(l,m,n) \tag{3}$$

Where $l$, $m$, $n$ are the dimensions of the 3D convolutional kernel, $M_i^{in}$ is the three-dimensional feature volume of the i-th channel of the input feature, and $W_j(l,m,n)$ represents the parameters of the j-th 3D convolutional kernel. The output feature after 3D convolution for the j-th channel is given by:

$$M_j^{out} = f(\sum_i m_{i,j}) \tag{4}$$

Where $f$ is the activation function.

### Decoupled 3D convolution neural network

Due to the high computational cost and the risk of overfitting associated with 3D CNN, we decompose it into two parts, referred to as decoupled 3D CNN. Experimental results indicate that decoupled 3D CNN outperforms 3D CNN in capturing the spatio-temporal correlations of traffic flow. The first part of decoupled 3D CNN establishes spatial correlations similar to 2D CNN, while the second part establishes temporal connections across frames. The integration of both parts achieves the same spatio-temporal receptive field as 3D CNN but significantly reduces computational complexity. As illustrated in **Fig 8(A)**, using a filter of size T×N×N for 3D CNN, where T is the temporal dimension, and N is the length and width of the spatial dimension. To reduce computational complexity, decoupled 3D CNN performs two consecutive convolution operations with filters of size 1×N×N and T×1×1, as shown in **Fig 8 (B)**. The computation process is as follows:

$$X_{3D1} = Conv3D_{(1,N,N)}(X_{3D}) \qquad (5)$$

$$X_{3D2} = Conv3D_{(T,1,1)}(X_{3D1}) \qquad (6)$$

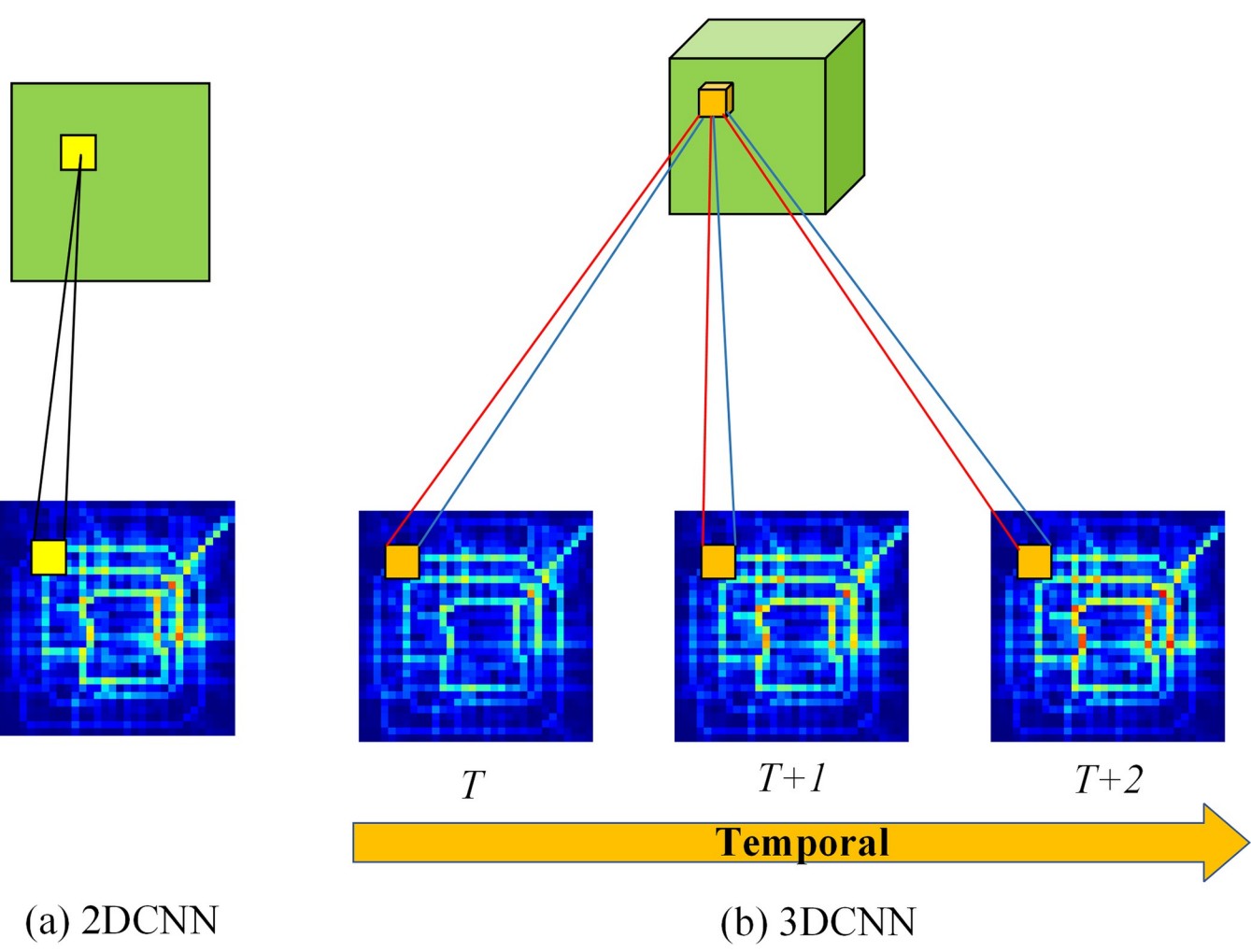

$T$ $T+1$ $T+2$

**Temporal**

(a) 2DCNN (b) 3DCNN

**Fig 7. Comparison between 2D CNN and 3D CNN.**

### Decoupled 3D DenseNet(D3DD)

The principle of DenseNet [36–39] is to use the outputs of all previous layers as inputs for the next layer, enhancing the data flow by obtaining features from multiple levels. Compared to ResNet, DenseNet can acquire more feature maps with fewer filters, which can reduce the number of parameters to some extent. Therefore, in this paper, we adopt densely connected decoupled 3D CNN to capture multi-level features of traffic flow, extract multiscale spatio-temporal dependencies, as illustrated in **Fig 9**. However, DenseNet has some drawbacks: its structure is not conducive to increasing the number of network layers. As the number of layers increases, the parameter count also sharply rises, leading to increased training difficulty. The computation process of the D3DD module is as follows:

$$X^L = F(X^1 \oplus X^2 \oplus \cdots X^{L-1}), L = 1, 2, 3, \ldots, l \tag{7}$$

Where $F$ represents the operation of decoupled 3D CNN, $\oplus$ denotes the concatenation operation, and $X^L$ represents the output of the L-th layer of the D3DD module.

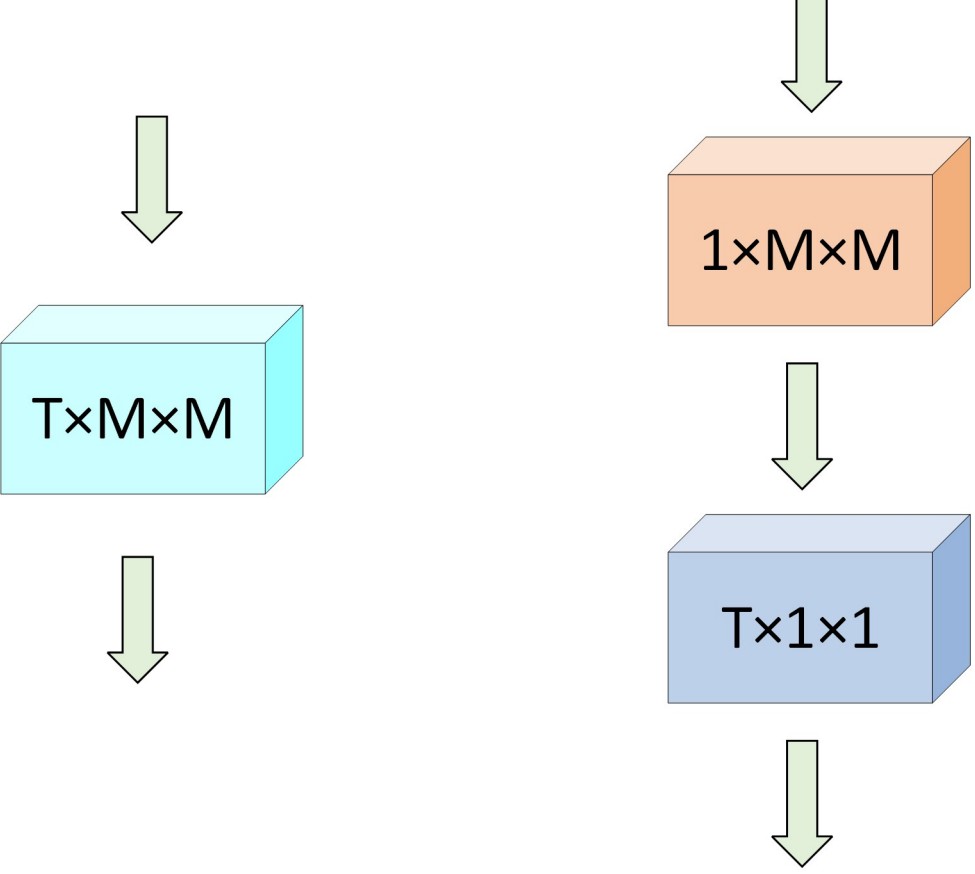

(a) 3D CNN          (b) Decoupled 3D CNN

**Fig 8. Comparison between ordinary 3D CNN and decoupled 3D CNN.**

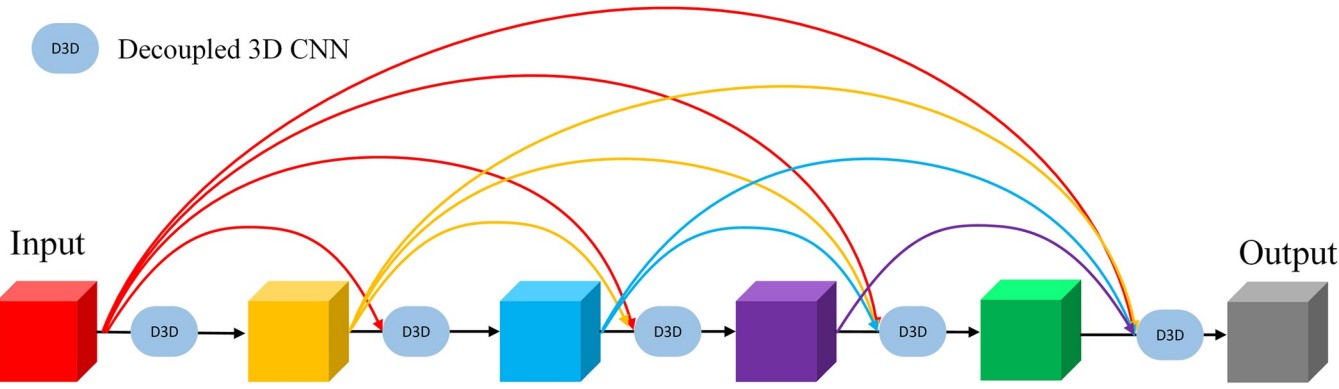

**Fig 9. Decoupled 3D denseNet architecture.**

## Attention residual network (ARN)

Due to the dynamic global spatial correlation in urban traffic flow, CNNs are constrained by the size of their convolutional kernels and cannot capture information over a large spatial range. Additionally, due to fixed parameters, these models lack the ability to dynamically capture data features, preventing them from fully learning the inherent information in the data. To dynamically explore the global spatial correlation of urban traffic flow and calculate the contribution of different channel features, we propose an Attention Residual Network (ARN), as illustrated in **Fig 10**. The network consists of multiple ARN Units connected by residual connections. Each ARN Unit comprises 2D CNN, spatial self-attention, and coordinate attention. The operation of residual connections is as follows:

$$X_A^L = X_A^{L-1} + f(X_A^{L-1}; \omega^{L-1}), L = 1, \cdots, l \tag{8}$$

Where $f$ represents all operations within the ARN Unit, $X_A^L$ represents the output of the L-th layer of the ARN module. and $\omega^{L-1}$ denotes all learnable parameters of the L-th ARN unit.

**Spatial self-attention mechanism.** The spatial self-attention structure, as shown in **Fig 11**, takes $X_S \in \mathbb{R}^{C \times H \times W}$ as input, where C being the number of channels, and the shape is initially transformed to $\mathbb{R}^{C \times N}$, where $N = H \times W$. Next, three fully connected layers are used to map from N-dimension to N-dimension separately, resulting in $Q_S \in \mathbb{R}^{C \times N}, K_S \in \mathbb{R}^{C \times N}, V_S \in \mathbb{R}^{C \times N}$. Next, calculate the attention result $X_{spatial} \in \mathbb{R}^{C \times N}$, and

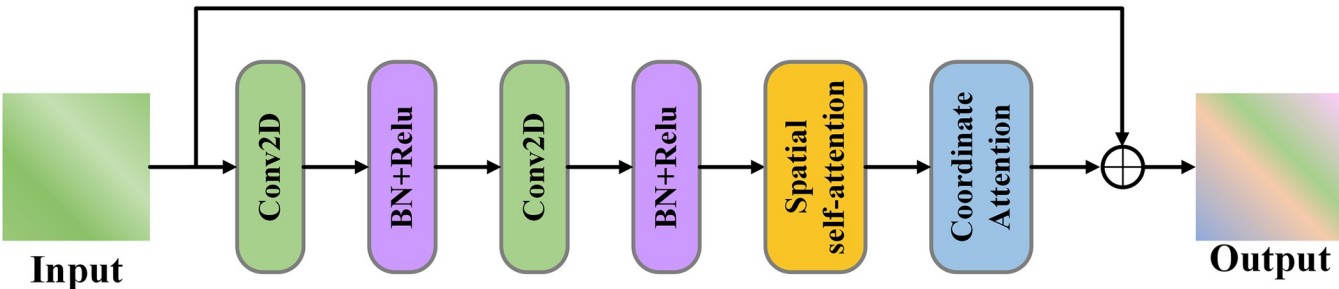

**Fig 10. ARN unit internal structure.**

finally reshape it to $\mathbb{R}^{C \times H \times W}$. The calculation process is as follows:

$$Q_S = W_{QS}X_S \tag{9}$$

$$K_S = W_{KS}X_S \tag{10}$$

$$V_S = W_{VS}X_S \tag{11}$$

$$X_{spatial} = softmax\left(\frac{Q_S K_S^T}{\sqrt{d_k}}\right)V_S \tag{12}$$

Where $W_{QS}, W_{KS}, W_{VS}$ are the parameter of the full connection layer, $d_K$ denote the dimension of $K_S$.

**Coordinate attention mechanism.** The structure of coordinate attention is shown in **Fig 12**. The input for coordinate attention is $X_{spatial} = \{M_1, M_2 \cdots M_c\} \in \mathbb{R}^{C \times H \times W}$, where $X_C \in \mathbb{R}^{H \times W}$ represents the feature of the c-th channel of $X_{spatial}$. The input features are pooled along both the horizontal and vertical directions. For the horizontal direction, average pooling is performed with a window size of $(1, W)$, and the output at position h for c-th channel is given by Eq (13). Similarly, for the vertical direction, average pooling with a window size of $(1, H)$ is applied, and the output at position $w$ for c-th channel is expressed by Eq (14).

$$z_c^h(h) = \frac{1}{W}\sum_{i=1}^{W} X_c(h, i) \tag{13}$$

Where $M_C(h,i)$ represents the value at the coordinate $(h,i)$ for c-th channel, $i = 1,2,3 \cdots W$.

$$z_c^w(w) = \frac{1}{H}\sum_{j=1}^{H} X_c(j, w) \tag{14}$$

Where $X_C(j,w)$ represents the value at the coordinate $(j,w)$ for c-th channel, $j = 1,2,3 \cdots H$.

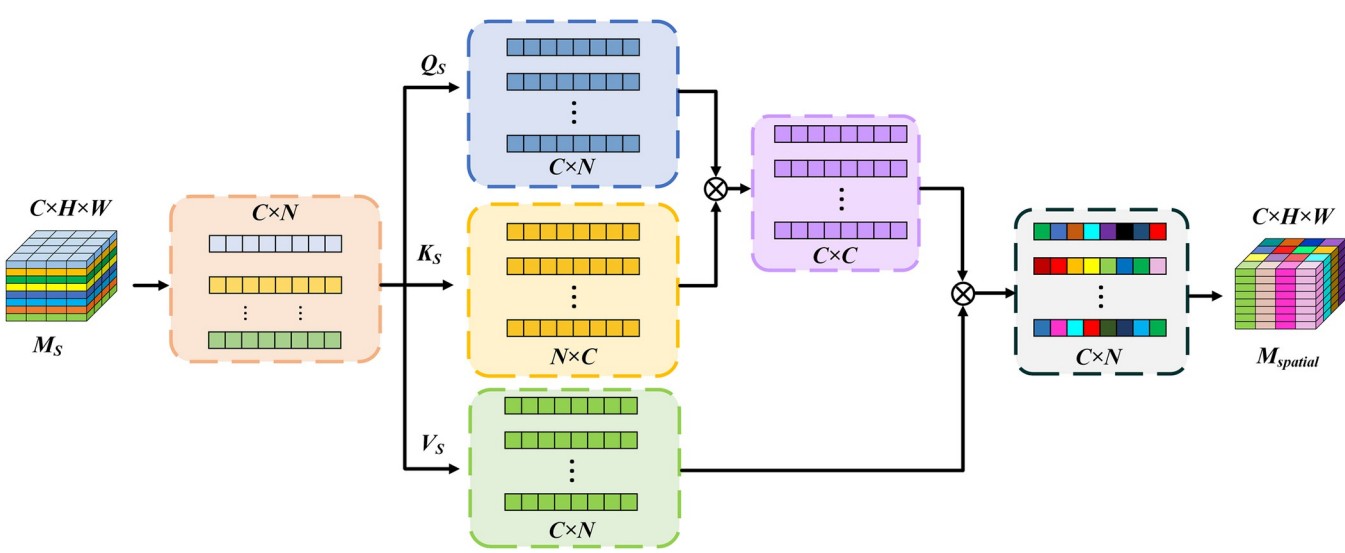

**Fig 11. Spatial self-attention structure.**

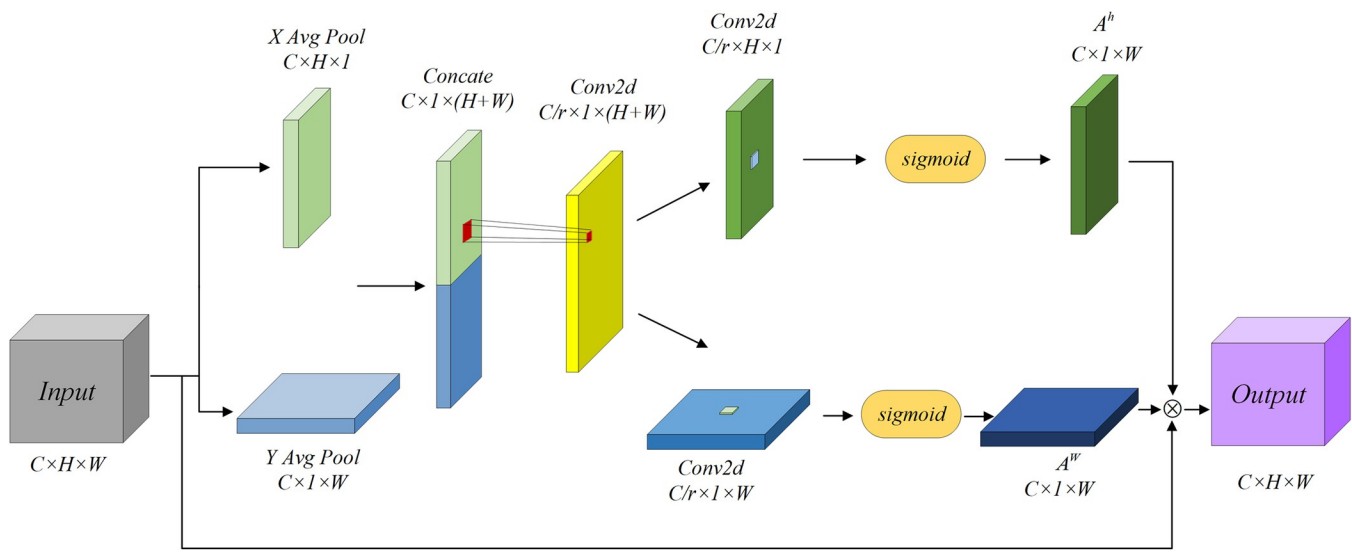

**Fig 12. Coordinate attention mechanism structure.**

Combine the pooled features from both horizontal and vertical directions, reduce the channel dimension to C/R through a 1x1 convolution operation, and finally apply the *ReLu* activation function to achieve thorough fusion of cross-channel spatial information. The process is as follows:

$$F = relu(Conv_{1\times1}([z^H, z^W])) \tag{15}$$

Where, $Z^H = \{z_1^H, z_2^H, z_3^H \cdots z_C^H\} \in \mathbb{R}^{C \times H \times 1}$, $Z^W = \{Z_1^W, Z_2^W, Z_3^W \cdots Z_C^W\} \in \mathbb{R}^{C \times 1 \times W}$, $F \in \mathbb{R}^{C/R \times 1 \times (W+H)}$.

Next, the interacted feature $F$ is separated into two independent features $F_h$ and $F_w$ based on their original sizes. Through a 1x1 convolution operation, the dimensionality of these features is restored, resulting in attention weights $A_h$ and $A_w$. The process is as follows:

$$A^h = sigmoid(Conv_{1\times1}(F_h)) \tag{16}$$

$$A^w = sigmoid(Conv_{1\times1}(F_w)) \tag{17}$$

Finally, the attention weight values $A_h, A_w$, and the input feature $X_{spatial}$ are multiplied to obtain the output $Y$. The process is as follows:

$$Y = X_{spatial} \otimes A^h \otimes A^w \tag{18}$$

**External factors branch.** The prediction of urban traffic flow is influenced by numerous external factors, thus incorporating these factors into the model can enhance the accuracy of traffic flow prediction. However, most existing methods only consider external factors at the predicted time, neglecting their continuous impact on traffic flow. For instance, after a rainstorm, the traffic flow does not immediately return to normal levels due to road flooding caused by heavy rain that continues to affect traffic. To address this issue, we incorporate multiple consecutive periods of external factors and capture their temporal information using long short-term memory networks (LSTMs). Subsequently, we map these features onto a traffic flow matrix shape through fully connected layers. The branch structure representing the incorporation of external factors is illustrated in **Fig 13**.

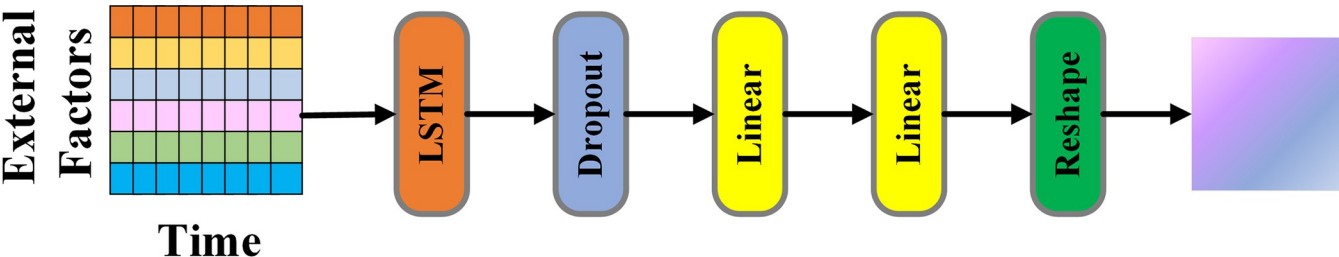

**Fig 13. External information branch structure.**

## Method implementation steps

The training process of ST-D3DDARN follows Algorithm 1, with inputs including the closeness, period, trend and external factors of urban traffic flow. The model extracts historical traffic flow features to predict the inflow and outflow for future time periods, transforming the problem into a regression task.

In Algorithm 1, $N$,$E_{ext}$, represent the inputs of the ST-D3DDARN model, and $\psi_{train}$ denotes the training set, where training instances are constructed from traffic data. Subsequently, the model undergoes training using backpropagation.

```
Algorithm 1: Training Process of the ST-D3DDARN Model
Input:
Given historical observations of urban traffic flow based on region:
N = {M¹,M²,M³,···,Mᵀ}
External factors: E_ext = {E¹_ext,E²_ext,E³_ext,···,Eᵀ_ext}
Closeness, period, trend and external factor sequence lengths: c, d,
w, l.
Output: ST-D3DDARN Model
1: ψ_train←∅
2: for t∈T do // T denotes all available time intervals
3: X_C = [M^{t-c},M^{t-(c-1)},M^{t-(c-2)},···,M^{t-1}]
4: X_P = [M_P^{t-l_d·d_daily},M_P^{t-(l_d-1)d_daily}···,M_P^{t-d_daily}], //d_daily represents the number of time
intervals within a day.
5: X_T = [M_T^{t-l_w·w_weekly},M_T^{t-(l_w-1)w_weekly}···,M_T^{t-w_weekly}] //w_weekly represents the number of
time intervals within a week.
6: E_ext = [E_ext^{t-l_e},E_ext^{t-(l_e-1)},E_ext^{t-(l_e-2)},···,E_ext^{t-1}]
7: Input the features X_C,X_P,X_T,E_ext into the corresponding branches of
the ST-D3DDARN model.
8: end for
9: // training this model
10: Initialize the parameters φ of the ST-D3DDARN model.
11: repeat
12: Select a batch of instances ψ_train randomly from ψ_batch
13: Through ψ_batch minimize loss function to find optimal φ
14 Until The maximum Epoch is reached or the early stopping condition
is satisfied
15 Return the trained ST-D3DDARN model
```

**Fig 14** illustrates the prediction process for the overall urban traffic flow. Initially, trajectory data collected by GPS sensors are processed into traffic frames. Subsequently, these traffic frames are divided into closeness time slices, period time slices and trend time slices, along with external factors, are organized into training, validation, and test sets. The ST-D3DDARN model is then constructed, and the training and validation sets are input into the model for training. Finally, the test set is fed into the trained model to validate the experimental results.

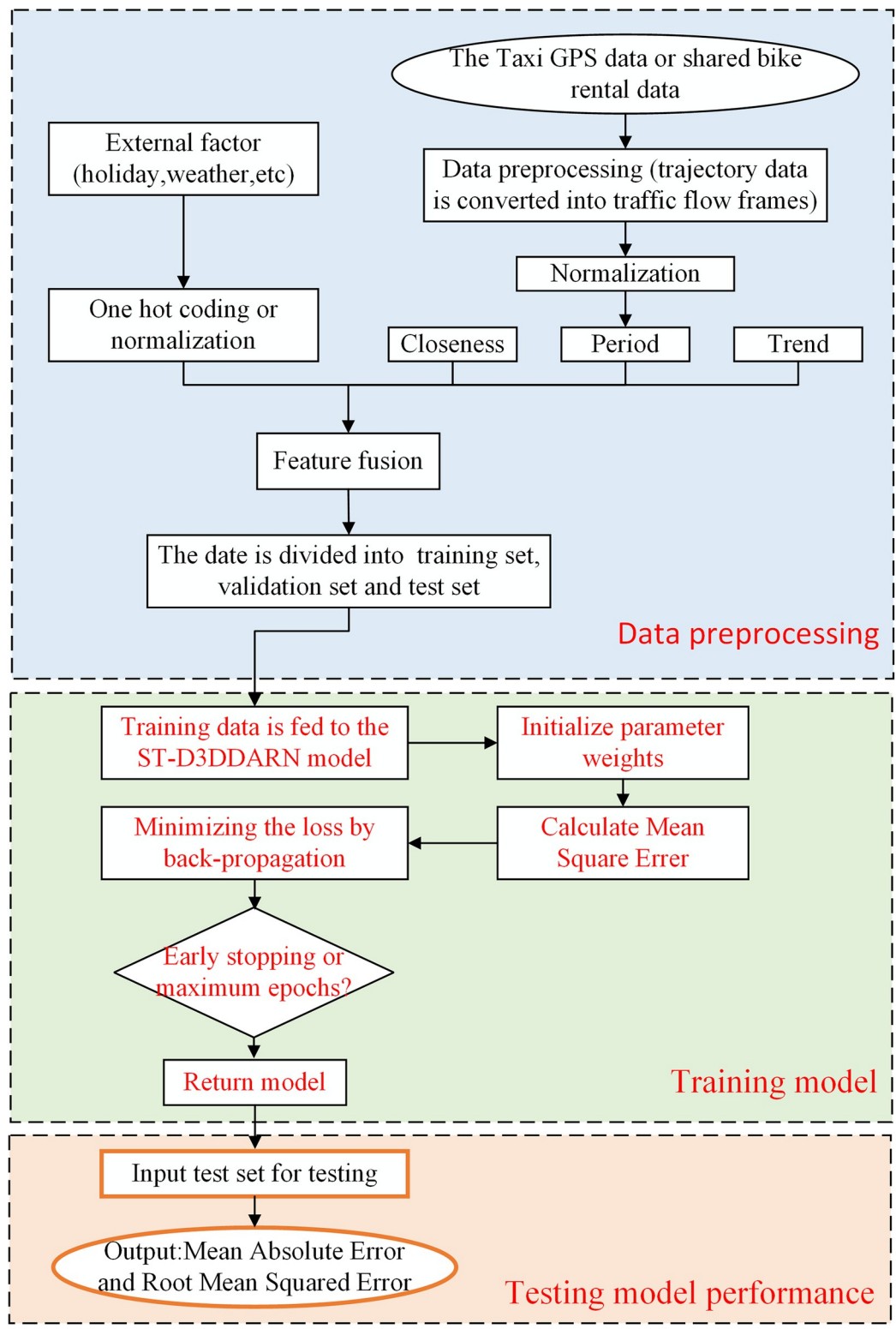

**Fig 14. Flow chart of traffic flow prediction in the whole city.**

**Table 1. Environment configuration.**

| Element | Parameter |
|---|---|
| Operating system | Windows10 |
| CPU | i9-12900K 3.20GHz |
| Memory | 64G |
| GPU | NVIDIA GeForce RTX 3090 |
| Cuda version | 12.2 |
| Language | Python3.9 |

## Experiment results and discussion

In this section, we begin by introducing the experimental environment and the dataset used. Subsequently, we analyze the impact of hyperparameters on the model and conduct comprehensive comparisons between the proposed model and various baseline models. Finally, to validate the effectiveness of each module, we perform ablation experiments. Additionally, to assess the model's effectiveness in multi-step prediction, we conduct performance tests for both direct multi-step prediction and recursive multi-step prediction.

### Experimental environment and experimental data

The experiment utilized PyTorch (version 1.13.1) to construct the ST-D3DDARN model. The specific experimental environment is outlined in **Table 1**.

To achieve optimal performance for each model, grid search was employed to fine-tune the hyperparameters of the baseline models. Each method underwent 10 experiments, and the average results were considered as references for model performance.

To validate the effectiveness of the proposed model, two representative trajectory datasets were utilized: the Beijing Taxi Trajectory dataset (TaxiBJ) and the New York City Bike Sharing dataset (BikeNYC). Detailed descriptions of the datasets are provided in

Table 2. Description of **TaxiBJ and BikeNYC**. The step lengths for closeness, period, trend and external factors were set to 6, 1, 1 and 6, respectively. The train-validate-test split ratio was 8:1:1, with a batch size of 32, learning rate of 0.005, and maximum training epochs of 200 and 100 for the two datasets. The loss function used for both datasets was the mean squared error (MSE) between actual and predicted traffic flows, calculated as follows:

$$loss = \|M^t - \hat{M}^t\|_2^2 \tag{19}$$

Where $M^t$ represents the actual traffic flow, and $\hat{M}^t$ represents the predicted traffic flow.

### Evaluation metric

To assess the accuracy of urban traffic flow predictions, two commonly used evaluation metrics, Mean Absolute Error (MAE) and Root Mean Squared Error (RMSE), are employed. The calculation formulas are as follows:

$$MAE = \frac{1}{N}\sum_{i=1}^{N}|\hat{M}_i^t - M_i^t| \tag{20}$$

$$RMSE = \sqrt{\frac{1}{N}\sum_{i=1}^{N}(\hat{M}_i^t - M_i^t)^2} \tag{21}$$

**Table 2. Description of TaxiBJ and BikeNYC.**

| Dataset | | TaxiBJ | BikeNYC |
|---|---|---|---|
| **Traffic flow** | Data Type | Taxi GPS Data | Bike Rental Data |
| | Time-span | 2013.07.01–2013.10.30 | 2014.04.01–2014.09.30 |
| | | 2014.03.01–2014.06.30 | |
| | | 2015.03.01–2015.06.30 | |
| | | 2015.11.01–2016.04.10 | |
| | Time interval | 30min | 60min |
| | Map Size | (32,32) | (16,8) |
| | Number of timestamps | 22459 | 4392 |
| | Taxi/Bike quantity | 34000+ | 6800+ |
| **External factor** | Vacation | 41 | 20 |
| | Weather | 16 | / |
| | Temperature/˚C | [-24.6,41.1] | / |
| | Wind Velocity/mph | [0,48.6] | / |

Where $N$ is Total number of validation samples, $M^t$ represents the actual traffic flow, and $\hat{M}^t$ represents the predicted traffic flow.

## Hyperparameter impact analysis

To further assess the effectiveness of the ST-D3DDARN model, this section analyzes the hyperparameters of two key modules: the number of layers in the D3DD module and the number of layers in the ARN module.

**Analysis of impact of the number of D3DD module layers.** The dense connections in the D3DD module enable feature reuse, thus establishing multi-scale spatio-temporal dependencies. The number of layers is an important parameter to reflect the fine-grained spatio-temporal dependence. The more layers are, the more spatio-temporal correlation is established. Through experiments on the ST-D3DDARN model with 2 layers of ARN modules, the relationship between the number of layers in the D3DD module and model performance was explored. The experimental results are shown in **Fig 15**.

**Fig 15** illustrates that RMSE and MAE exhibit a pattern of initial decline followed by an increase as the number of D3DD module layers increases. This phenomenon can be attributed to the existence of multi-scale spatio-temporal correlations in urban traffic flows, which are captured more comprehensively with additional D3DD module layers. However, excessive use of these modules may introduce remote area effects that are irrelevant or weakly related to the target area. Furthermore, it is recommended to set the number of D3DD module layers at 4 and 2 for TaxiBJ and BikeNYC datasets respectively. This result proves that there are more scales of spatio-temporal correlation in the data set with larger city extent and more number ofgirds.

**Analysis of impact of the number of aRN module layers.** The number of layers in the Attention Residual Network (ARN) module is a crucial parameter that significantly impacts the performance of traffic flow prediction in urban areas. To thoroughly analyze the influence of ARN module layers on the performance of the ST-D3DDARN model, we conducted experiments by fixing the D3DD module layers at 4 and 2 for the TaxiBJ and BikeNYC datasets, respectively. We gradually increased the number of ARN module layers and observed the effects, as depicted in **Fig 16**.

From the observed trend in the graph, it is evident that with an increase in the number of ARN module layers, the model's performance initially improves and then tends to stabilize.

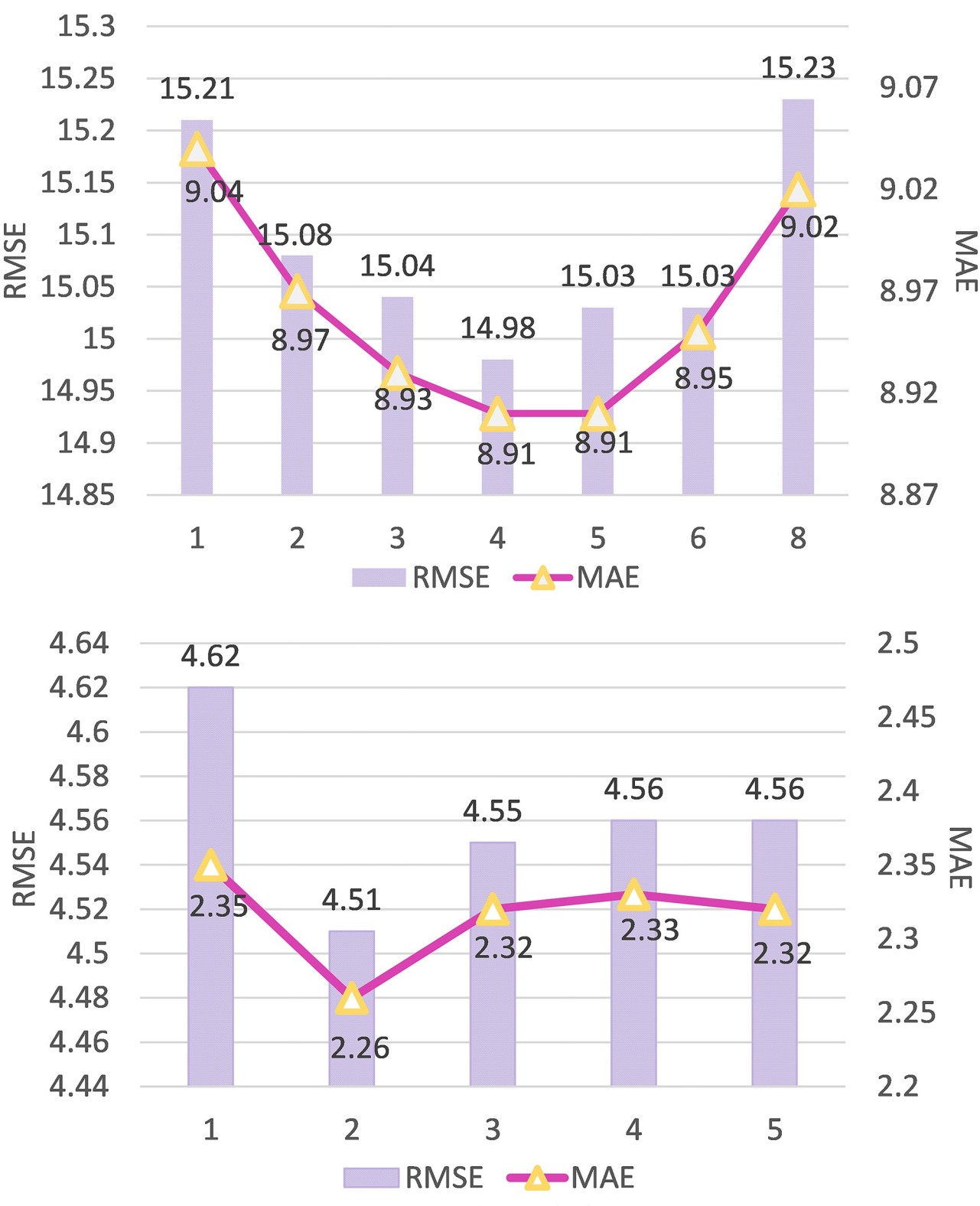

**Fig 15. Experimental results with different number of D3DDN layers.** (a) TaxiBJ. (b) BikeNYC.

This suggests that increasing the number of ARN module layers can enhance the model's ability to learn spatio-temporal correlations in urban traffic flow to a certain extent. However, beyond a certain threshold of layer count, the performance improvement becomes marginal, and there is a risk of introducing overfitting. Therefore, in practical applications, a trade-off between model performance and computational efficiency is necessary, and an appropriate number of ARN module layers should be selected. So we set the number of layers of ARN module to 2.

## Comparison of baseline methods

This section compares the ST-D3DDARN model with baseline models to demonstrate the effectiveness of the proposed approach.

**Introduction to baseline methods.** Traditional Time Series Forecasting Models:

- **HA:** Historical Average model calculates the historical average and combines it with the current traffic flow at the current time to predict future traffic flow;

- **ARIMA [17]:** AutoRegressive Integrated Moving Average model predicts future traffic flow trends based on the autocorrelation of historical traffic flow data;

- Deep Learning Models:

- **CNN+LSTM:** A combination model of temporal and spatial features using CNN and LSTM to extract time and space features of traffic flow asynchronously;

- **GCN+LSTM:** It is also a combination model of temporal and spatial. In this method, a graph G = (V, E) is constructed, where V and E represent the set of vertices and edges respectively. Each vertex in a graph G represents a region, and an edge between two vertices indicates that two regions are adjacent.

- **ST-ResNet [9]:** A deep neural network model for traffic prediction that captures spatio-temporal correlations separately from closeness, periodicity, and trend;

- **DeepSTN+ [11]:** A deep neural network model that integrates closeness, period, and trend, followed by ResPlus units to fuse multi-scale features of traffic flow;

- **ST-3DNet [25]:** The first model to use 3D convolution to capture spatio-temporal correlations in traffic flow, showing superior performance on bikeNYC and TaxiBJ;

- **3D-ConvLSTMNet [28]:** Captures short-term spatio-temporal correlations with 3D CNN, followed by ConvLSTM to capture long-term spatio-temporal correlations, and utilizes residual connections for long-distance spatial dependencies;

- **MS-ResCnet [13]:** A multi-scale residual calibration network that fuses multi-scale spatio-temporal features through deep interleaved training;

- **MPCNN [14]:** A multi-perspective convolutional network that convolves features from the top, front, and side perspectives to capture richer features.

Table 3 provides a brief comparison of ST-D3DDARN with various baseline models, helping to illustrate the differences and enhance the interpretability of each model.

**Comparison with baseline performance.** The experimental results of each method are presented in **Tables 4 and 5**. In the baseline model, classical time forecasting methods (ARIMA and HA) exhibit limited effectiveness as they solely rely on historical values for future predictions, disregarding spatio-temporal correlation and external factors. Deep learning methods encompass CNN+LSTM, GCN+LSTM, ST-ResNet, DeepSTN+, ST-3DNet, 3DConvLSTMNet, MS-ResCnet. CNN+LSTM and GCN+LSTM capture the spatio-temporal

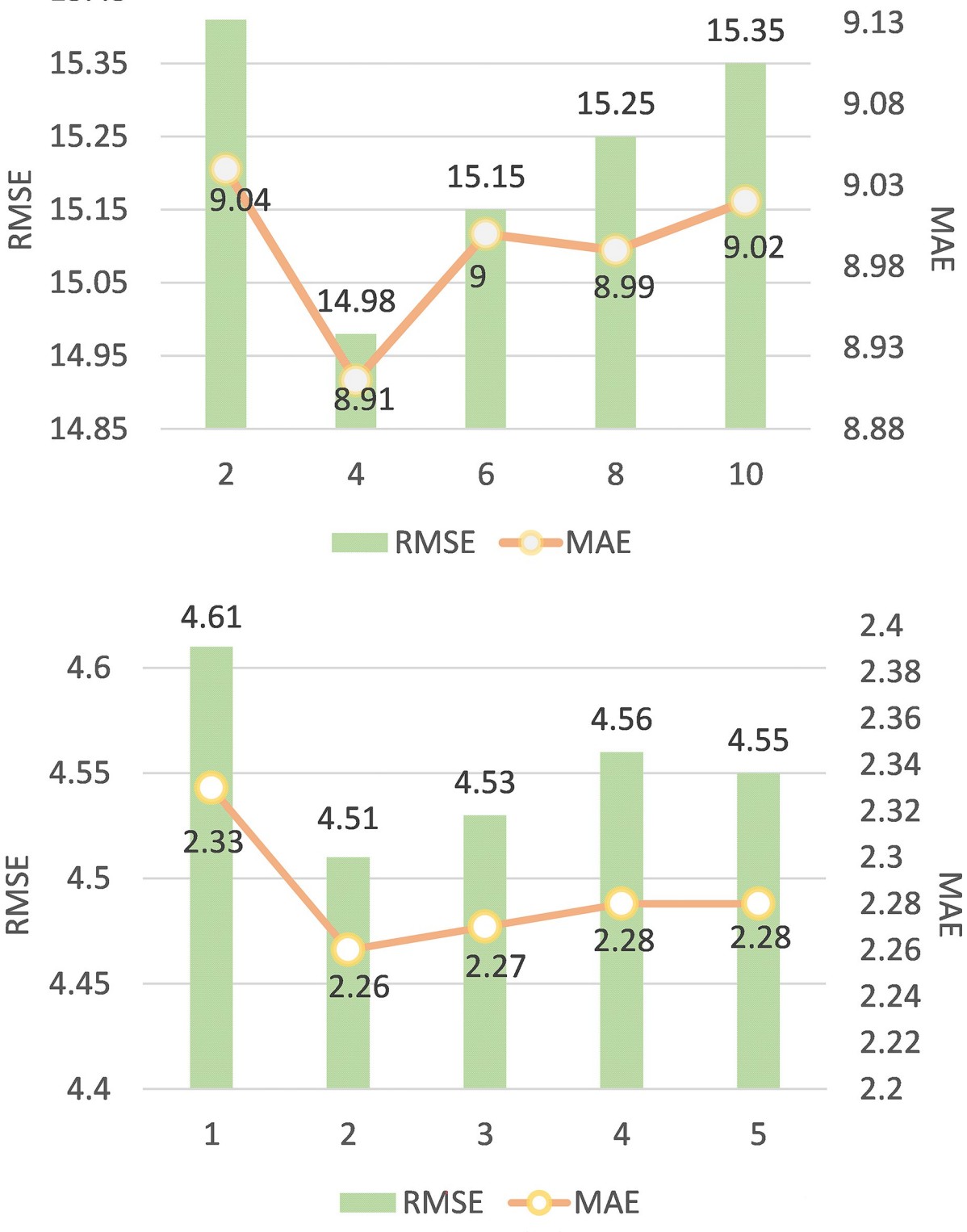

**Fig 16. Experimental results of different ARN module layers.** (a) TaxiBJ. (b) BikeNYC.

**Table 3. Comparison of model characteristics.**

| Methods | Temporal [1] | Spatial [2] | Spatio-temporal [3] | Global spatial [4] | Multiscale [5] | Heterogeneity [6] |
|---|---|---|---|---|---|---|
| HA | ✓ | | | | | |
| ARIMA | ✓ | | | | | |
| CNN+LSTM | ✓ | ✓ | | | | |
| GCN+LSTM | ✓ | ✓ | | | | |
| ST-ResNet | ✓ | ✓ | | | | |
| DeepSTN+ | ✓ | | | ✓ | | |
| ST-3DNet | ✓ | ✓ | ✓ | | | ✓ |
| 3D-convLSTMNet | ✓ | ✓ | ✓ | | | |
| MS-ResCnet | ✓ | ✓ | | | ✓ | |
| MPCNN | ✓ | ✓ | ✓ | | | |
| **ST-D3DDARN** | ✓ | ✓ | ✓ | ✓ | ✓ | ✓ |

[1] Temporal correlation of traffic flow

[2] Spatial correlation of traffic flow

[3] Spatio-temporal dependence of traffic flow

[4] Global spatial correlation

[5] Spatio-temporal correlations at multiple scales

[6] Heterogeneity of traffic flow.

correlation of urban traffic in a spatio-temporal asynchronous manner, it overlooks traffic flow periodicity resulting in subpar prediction outcomes. ST-ResNet establishes closeness, period, and trend branches, leading to improved accuracy compared to traditional models. DeepSTN+ achieves remarkable results by establishing citywide spatial dependencies through ResPlus units. However, ResPlus units incur high computational costs when dealing with data-sets with many grids. ST-3DNet simultaneously captures traffic flow's spatio-temporal correlation via 3D convolution while proposing a recalibration module that explicitly quantifies the contribution difference of spatial correlation, effectively addressing traffic flow heterogeneity. 3D-ConvLSTMNet outperforms ST-3DNet by capturing long-term dependencies through the ConvLSTM architecture; Nevertheless, both methods fall short of expectations on the two datasets, indicating ordinary 3D CNN is unsuitable for capturing traffic flow's spatio-temporal characteristics adequately. MPCNN establishes traffic flow's spatio-temporal correlation from

**Table 4. Comparison of model prediction results.**

| Methods | TaxiBJ | | BikeNYC | |
|---|---|---|---|---|
| | RMSE | MAE | RMSE | MAE |
| HA | 45.36 | 22.47 | 20.33 | 10.22 |
| ARIMA | 22.78 | 12.72 | 10.22 | 5.36 |
| CNN+LSTM | 17.29 | 9.87 | 5.48 | 2.83 |
| GCN+LSTM | 16.79 | 9.84 | 5.28 | 2.73 |
| ST-ResNet | 16.92 | 9.64 | 5.27 | 2.75 |
| DeepSTN+ | 16.29 | 9.27 | 4.66 | 2.43 |
| ST-3DNet | 16.73 | 9.69 | 5.22 | 2.62 |
| 3D-ConvLSTMNet | 16.47 | 9.58 | 5.14 | 2.96 |
| MS-ResCnet | 15.60 | 9.15 | 4.82 | 2.48 |
| MPCNN | 15.77 | 9.23 | 5.02 | 2.47 |
| **ST-D3DDARN** | **14.98** | **8.91** | **4.49** | **2.25** |

**Table 5. Inflow and outflow prediction results in TaxiBJ.**

| Methods | Inflow | | Outflow | |
|---|---|---|---|---|
| | RMSE | MAE | RMSE | MAE |
| HA | 45.27 | 22.40 | 45.45 | 22.54 |
| ARIMA | 22.73 | 12.56 | 22.82 | 12.83 |
| CNN+LSTM | 17.27 | 9.85 | 17.32 | 9.90 |
| GCN+LSTM | 16.75 | 9.82 | 16.83 | 9.86 |
| ST-ResNet | 16.89 | 9.62 | 16.95 | 9.66 |
| DeepSTN+ | 16.26 | 9.24 | 16.32 | 9.30 |
| ST-3DNet | 16.67 | 9.66 | 16.78 | 9.74 |
| 3D-ConvLSTMNet | 16.39 | 9.52 | 16.55 | 9.64 |
| MS-ResCnet | 15.56 | 9.08 | 15.64 | 9.15 |
| MPCNN | 15.73 | 9.19 | 15.81 | 9.26 |
| **ST-D3DDARN** | **14.86** | **8.81** | **15.11** | **8.99** |

various perspectives, thereby enhancing prediction accuracy to some extent. MS-ResCnet uses a two-channel ResCnet network method to extract the benchmark features and calibration features of traffic flow respectively, and combines multi-scale features to further improve the prediction accuracy. It shows that extracting multi-scale features of traffic flow can effectively improve model prediction performance. The proposed ST-D3DDARN model integrates the strengths of each model, and the decoupled 3D CNN effectively addresses the training challenges of conventional 3D CNN. Therefore, this model performs optimally on both datasets.

## Prediction error visualization

To showcase the predictive performance of each method, we visualized the prediction errors for three time periods on April 10, 2016 (TaxiBJ) and September 30, 2014 (BikeNYC), as shown in **Fig 17**. The error plots reveal that the proposed method exhibits better error control during peak traffic hours or specific regions compared to the baseline models.

## Efficiency evaluation

To assess the prediction efficiency of ST-D3DDARN, we compared its complexity with various baseline models, and the details of the comparisons are provided in **Table 6** and **Fig 18**. Due to the complexity of the parameters in each model, it is challenging to display their configurations one by one. For detailed parameters, please refer to the original papers of each baseline model or the source code provided in this study.

Among the baseline models, DeepSTN+ exhibits the fastest convergence speed, but its embedding of fully connected layers within the ResPlus unit introduces a large number of parameters. ST-3DNet and 3D-ConvLSTMNet, utilizing standard 3D CNN, result in slow training speeds and poor convergence. Comparatively, MS-ResCnet and MPCNN have simpler structures, smaller parameter counts, and faster convergence speeds. The spatial self-attention mechanism in ST-D3DDARN introduces a certain number of parameters, fortunately, the computational load generated by these parameters is acceptable. Overall, whether considering prediction accuracy or model efficiency, ST-D3DDARN demonstrates certain advantages.

## Ablation experiment

This section presents the ablation experiments of ST-D3DDARN to analyze the impact of each module on model performance. Due to space constraints, not all variant combinations are

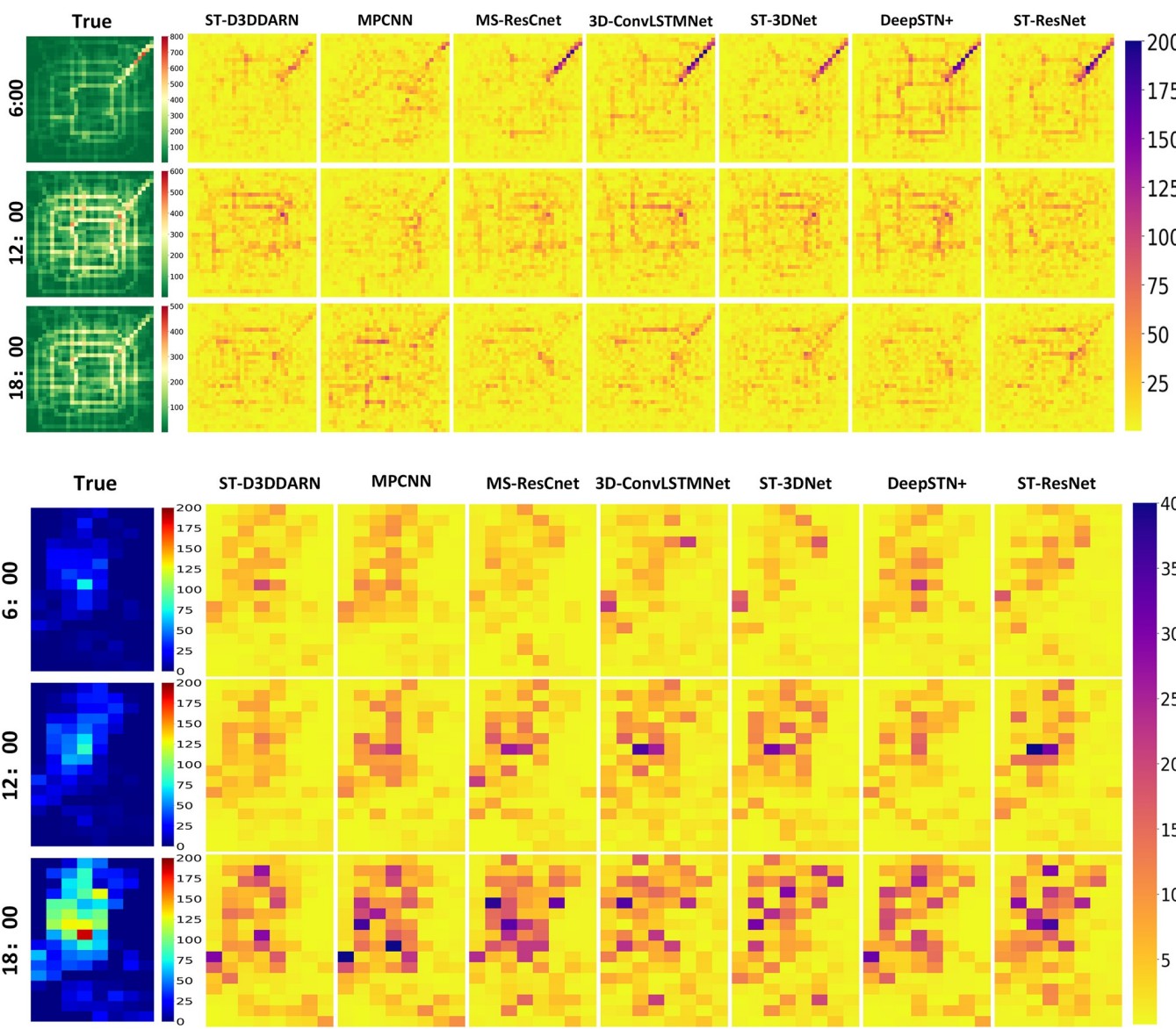

**Fig 17. Error comparison of different models.** (a) TaxiBJ. (b) BikeNYC.

Table 6. Comparison of model efficiency.

| Methods | Parameters(M) | Time each Epoch(s) | Testing time(ms) | Epoch to converge | Video memory footprint(GB) |
|---|---|---|---|---|---|
| ST-ResNet | 0.96 | 15.96 | 428.77 | 88 | 3.19 |
| 3D-ConvLSTMNet | 1.44 | 17.35 | 587.04 | 99 | 3.83 |
| DeepSTN+ | 270.8 | 36.4 | 601.99 | 45 | 8.28 |
| ST-3DNet | 1.20 | 14.58 | 415.61 | 175 | 3.67 |
| MS-ResCnet | 0.51 | 10.4 | 338.87 | 73 | 3.56 |
| MPCNN | 0.89 | 10.09 | 406.65 | 72 | 3.32 |
| **ST-D3DDARN** | **1.30** | **7.06** | **447.92** | **70** | **3.05** |

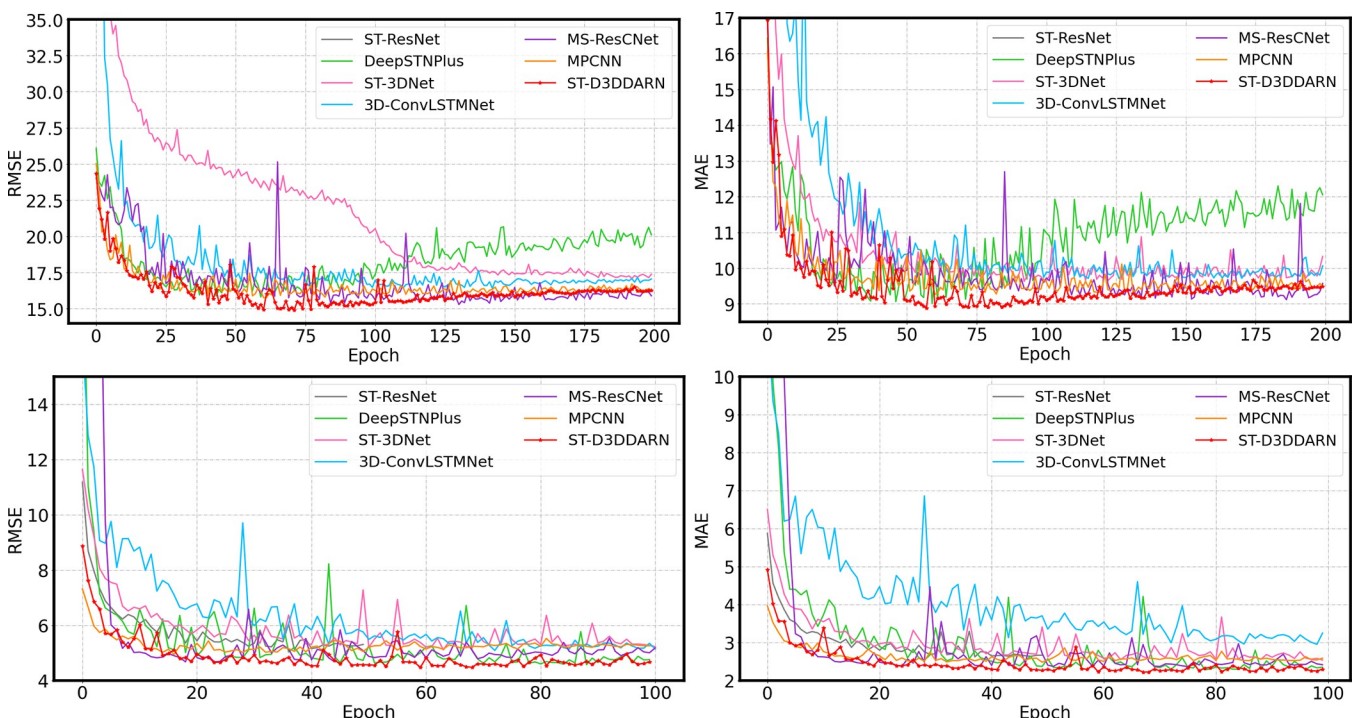

**Fig 18. RMSE and MAE changes of the test set during the training of each model.** (a) TaxiBJ of RMSE. (b) TaxiBJ of MAE. (c) BikeNYC of MAE. (d) BikeNYC of MAE.

included in this study, but rather aims to verify the effectiveness of each module. Specifically, we analyzed six variants:

- ST-D3DDARN-w/o-CA: This variant removes coordinate attention from the model;

- ST-D3DDARN-w/o-SA: This variant removes spatial self-attention from the model;

- ST-D3DDARN-w/o-D3DD: This variant removes the decoupled 3D CNN with dense connections in the model;

- ST-D3DDARN-3D: This variant replaces the decoupled 3D CNN in the model with a normal 3D CNN;

- ST-D3DDARN-w/o-EXT: This variant removes the external factor branch in the model;

- ST-D3DDARN-w/o-ADD. This variant is not supplemented with additional traffic flow time slices.

**Table 7** shows various indicators for our proposed model and its variants in TaxiBJ. Our findings indicate that both coordinate attention and spatial self-attention have an impact on prediction accuracy; however, spatial self-attention has a more significant influence which highlights the importance of dynamically capturing global spatial correlation. The prediction accuracy decreases when using ordinary 3D CNN instead of decoupled 3D CNN with dense connections indicating that it is unsuitable for capturing spatio-temporal correlation in traffic flow. Results from experiments without external factors show that adding them improves traffic flow prediction while additional traffic frames are necessary to deal with translation phenomena.

**Table 7. Comparison of model variants of TaxiBJ.**

| Variants | RMSE | MAE | Parameters(M) | Time each Epoch(s) |
|---|---|---|---|---|
| ST-D3DDARN-w/o-CA | 15.21 | 9.05 | 1.29M | 6.38 |
| ST-D3DDARN-w/o-SA | 15.43 | 9.16 | 0.24M | 6.17 |
| ST-D3DDARN-w/o-D3DD | 15.25 | 9.06 | 1.29M | 4.58 |
| ST-D3DDARN-3D | 15.32 | 9.20 | 1.31M | 7.33 |
| ST-D3DDARN-w/o-EXT | 15.29 | 9.11 | 1.27M | 6.86 |
| ST-D3DDARN-w/o-add | 15.39 | 9.05 | 1.30M | 7.02 |
| **ST-D3DDARN** | **14.98** | **8.91** | **1.30M** | **7.06** |

## Multistep prediction

Multi-step prediction is more practically significant than single-step prediction. In this section, we analyze the multi-step prediction results of each method.

**Direct multi-step prediction.** As shown in **Fig 19**, our proposed method consistently outperforms other baselines at all steps on both datasets. This is attributed to our proposed method effectively capturing the spatio-temporal correlations in traffic flow. However, direct multi-step prediction involves building multiple models for each step, which can lead to a heavy computational and maintenance burden when predicting a large number of time steps.

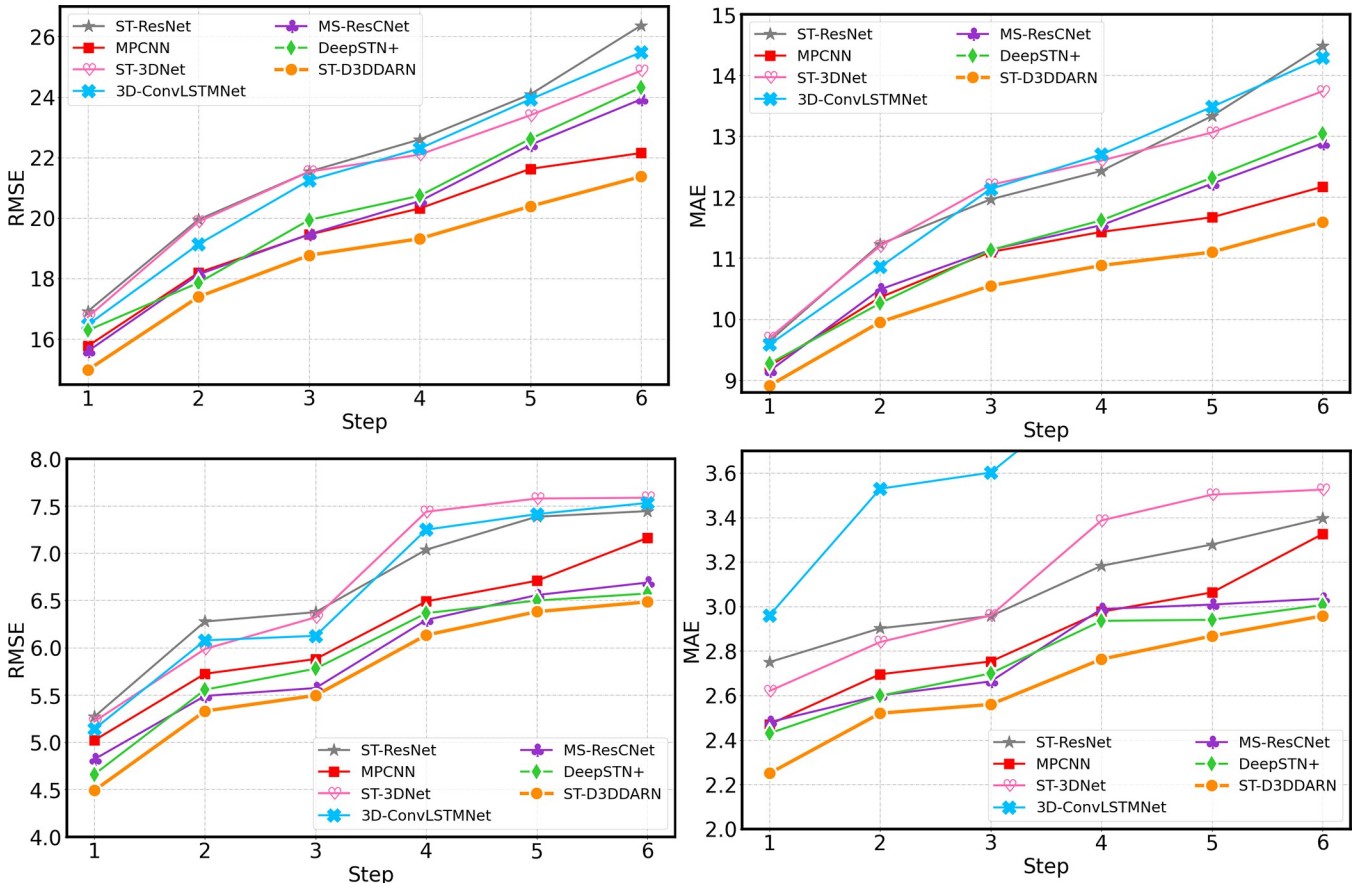

**Fig 19. Performance of direct multi-step predictions for each method.** (a) Step-wise RMSE of TaxiBJ. (b) Step-wise MAE of TaxiBJ. (c) Step-wise RMSE of BikeNYC. (d) Step-wise MAE of BikeNYC.

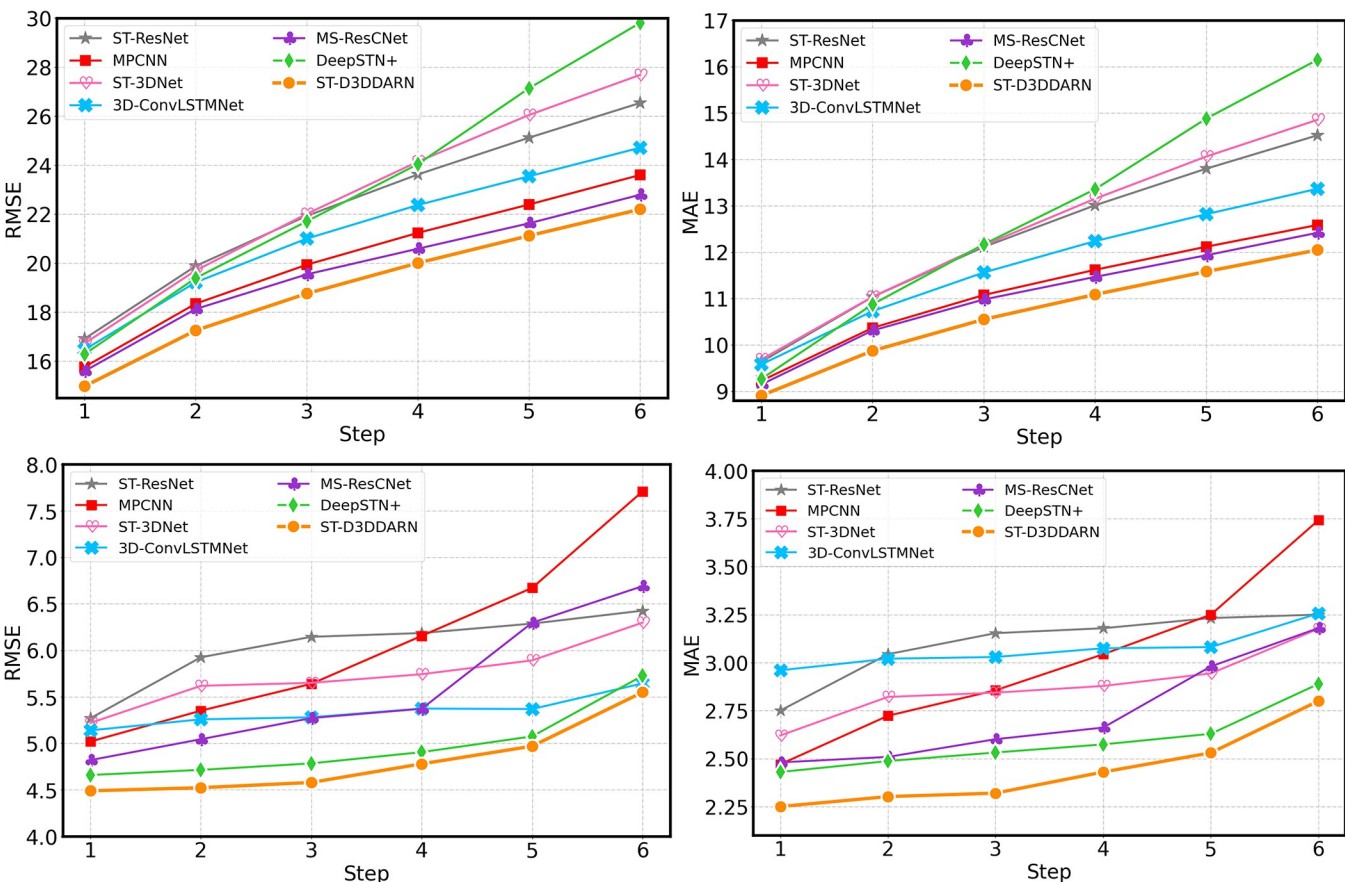

**Fig 20. Performance of recursive multi-step predictions for each method.** (a) Step-wise RMSE of TaxiBJ. (b) Step-wise MAE of TaxiBJ. (c) Step-wise RMSE of BikeNYC. (d) Step-wise MAE of BikeNYC.

**Recursive multi-step prediction.** Compared with the direct multi-step prediction, the recursive multi-step prediction saves a lot of computing power by substituting the prediction result of the previous step into the model, so this method has strong practicability. The prediction results of the recursive multi-step prediction of each model are shown in **Fig 20**. The prediction accuracy of the recursive multi-step prediction of most models is not significantly decreased compared with the direct multi-step prediction, and the ST-D3DDARN model always shows high accuracy in each step prediction in the two datasets.

## Conclusions

In order to comprehensively and efficiently extract the spatio-temporal characteristics of traffic flow, this paper proposes the ST-D3DDARN model for urban regional traffic flow prediction. In this model, the D3DD module is utilized to extract multi-scale spatio-temporal features at a lower level, while a residual network integrating spatial self-attention and coordinate attention is designed. The experiment proves that the decoupled 3D CNN is more suitable for extracting the spatio-temporal phase of traffic flow than the 3D CNN. Additionally, the spatial self-attention mechanism employed in this study effectively establishes dynamic spatial correlations within city traffic flows. At the same time, coordinate attention accurately captures contributions from different channel features. Furthermore, supplementary traffic frames are proven necessary through extensive experimentation. The results indicate that ST-D3DDARN

outperforms baseline models across all aspects evaluated. Notably, this model can be widely applied to predict spatio-temporal rasterized traffic data and provide reliable information for transportation management departments in intelligent transportation systems.

Future work may involve dividing cities into irregular regions based on their functional characteristics and modeling urban traffic flows using GCN to learn spatio-temporal information better. Moreover, incorporating functional area attributes (such as commercial district/residential data and POI data) into the model could further enhance its predictive performance.

## Acknowledgments

We thank the editors and the reviewers for their valuable comments and suggestions.

## Author Contributions

**Formal analysis:** Guowei Yang, Zhaochong Zhang.

**Methodology:** Jing Chen, Guowei Yang.

**Project administration:** Zhaochong Zhang, Wei Wang.

**Supervision:** Jing Chen.

**Visualization:** Guowei Yang.

**Writing – original draft:** Jing Chen, Guowei Yang.

**Writing – review & editing:** Jing Chen, Wei Wang.

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
