## [Decision Letter · Decision Letter 0]

5 Apr 2024

PONE-D-24-03623ST-D3DDARN: Urban Traffic Flow Prediction based on Spatio-temporal Decoupled 3D Densenet with Attention ResNetPLOS ONE

Dear Dr. Wei,

Thank you for submitting your manuscript to PLOS ONE. After careful consideration, we feel that it has merit but does not fully meet PLOS ONE’s publication criteria as it currently stands. Therefore, we invite you to submit a revised version of the manuscript that addresses the points raised during the review process.

We look forward to receiving your revised manuscript.

Kind regards,

Muhammad Bilal

Academic Editor

PLOS ONE

Journal Requirements:

4. Thank you for uploading your study's underlying data set. Unfortunately, the repository you have noted in your Data Availability statement does not qualify as an acceptable data repository according to PLOS's standards.

Reviewers' comments:

Reviewer's Responses to Questions

**Comments to the Author**

1. Is the manuscript technically sound, and do the data support the conclusions?

Reviewer #1: Yes

Reviewer #2: Yes

2. Has the statistical analysis been performed appropriately and rigorously? 

Reviewer #1: Yes

Reviewer #2: Yes

3. Have the authors made all data underlying the findings in their manuscript fully available?

Reviewer #1: Yes

Reviewer #2: Yes

4. Is the manuscript presented in an intelligible fashion and written in standard English?

Reviewer #1: Yes

Reviewer #2: Yes

5. Review Comments to the Author

**Reviewer #1**: This manuscript represents traffic flow as traffic frames inspired by video analysis in computer vision and proposes an end-to-end urban traffic flow prediction model, named Spatio-temporal Decoupled 3D Densenet with Attention ResNet (ST-D3DDARN). This model extracts multi-source traffic flow features and dynamically establishes global spatio-temporal correlations with attention residual network. The overall experiments in the manuscript are sufficient, I think there are still some areas that need revision. Specific suggestions are provided below：

1.Pay attention to some formatting in writing. There are many formatting issues with the manuscript. For example, the second paragraph in the introduction part should not be written in the top case. The title of Part 3 “Problem Definition and Analysis” lacks punctuation. Please carefully check.

2. There are multiple citation errors in the introduction section. [Error! Reference source not found.] appears repeatedly. Additionally, in the related work section, there are several repeated references, such as Lin et al. [1111], Pu et al. [3636]. Please resolve them.

3. Please note the title of the figure. In Analysis 3 of the Definition 4 , there are Fig1 and Fig 2, but the introduction part also includes the Fig 1 and Fig 2, which are not the same. In addition, Fig 3 refers to “Urban grid and definition of inflow and outflow”, as well as “The overall framework of the ST-D3DDARN model”, Please correct them.

4. The manuscript controls the parameters and efficiency of the model, where "Time each Epoch" in Table 6 refers to training time? If so, why is the model less time-consuming during training than all comparison models, but more time-consuming during testing? Actually, in practical use of the model, testing time is generally more important.

5. Dividing the transportation network into grids is an approximate method, does it affect prediction accuracy? There are many papers that use graph for traffic prediction, for example “Hierarchical Spatio-Temporal Graph Convolutional Networks and Transformer Network for Traffic Flow Forecasting”, “Knowledge fusion enhanced graph neural network for traffic flow prediction”, you can refer to them.

**Reviewer #2: **Overall, in this paper, the authors have made significant contributions to increase the knowledge. There are a few grammar things that could be improved. Could you double-check the grammar mistakes?

Abstract, introduction, contribution and conclusion should be aligned.

6. PLOS authors have the option to publish the peer review history of their article (what does this mean?). If published, this will include your full peer review and any attached files.

Reviewer #1: **Yes: **Yong Zhang

Reviewer #2: No

---

## [Author Response · Author response to Decision Letter 0]

13 May 2024

Response to editor: 

Response:Thank you for providing the links. We will ensure that our manuscript complies with PLOS ONE's style requirements, including file naming conventions. We will utilize the style templates provided by PLOS ONE for formatting the main body and the title, authors, and affiliations sections. When submitting the revised version, we will adhere to PLOS ONE's style guidelines to ensure our manuscript meets the journal's standards.

Response:Thank you for bringing this to our attention. We will review the guidelines provided by PLOS ONE on code sharing. We understand the importance of making all author-generated code freely available without restrictions upon publication to promote reproducibility and facilitate reuse of the research. We have already upload experimental code and relevant documentation to https://github.com/761049669/ST-D3DDARN.

Response:Thank you for bringing this to our attention. We will ensure that the grant information provided in both the 'Funding Information' and 'Financial Disclosure' sections match. In the resubmission, we will provide the correct grant numbers for the awards received for our study in the 'Funding Information' section to ensure accuracy and consistency.

4. Thank you for uploading your study's underlying data set. Unfortunately, the repository you have noted in your Data Availability statement does not qualify as an acceptable data repository according to PLOS's standards.

Response: Thank you for your remind, we have already upload the minimum data set used in experiments to https://github.com/761049669/ST-D3DDARN.

Response: Thank you for the reminder. As for the data availability statement issues involved in this paper, all authors have agreed to share all code and data sets freely at present. Our code and dataset can be found on the https://github.com/761049669/ST-D3DDARN.

Thank you again for your careful work on my manuscript and please contact me if you need any further information.

Reviewer #1

This manuscript represents traffic flow as traffic frames inspired by video analysis in computer vision and proposes an end-to-end urban traffic flow prediction model, named Spatio-temporal Decoupled 3D Densenet with Attention ResNet (ST-D3DDARN). This model extracts multi-source traffic flow features and dynamically establishes global spatio-temporal correlations with attention residual network. The overall experiments in the manuscript are sufficient, I think there are still some areas that need revision. Specific suggestions are provided below：

Response: Thanks for kindly reviewing our work. Please refer to the responses for our point-by-point replies.

Point 1: Pay attention to some formatting in writing. There are many formatting issues with the manuscript. For example, the second paragraph in the introduction part should not be written in the top case. The title of Part 3 “Problem Definition and Analysis” lacks punctuation. Please carefully check.

Response: We are sorry that our paper suffers from many formatting problems. We have carefully checked the paper for formatting issues and made the necessary changes. In the introduction, we changed the case form in several places. In addition, we modified the format of all titles according to the requirements of the journal. Thank you for your patience and we will ensure that the paper is in the right format.

Point 2: There are multiple citation errors in the introduction section. [Error! Reference source not found.] appears repeatedly. Additionally, in the related work section, there are several repeated references, such as Lin et al. [1111], Pu et al. [3636]. Please resolve them.

Response: Thank you for your careful review! We apologize for some minor mistakes that were inadvertently made in the text. We carefully checked the citations in the paper and fixed the citation errors that occurred in the introduction section, ensuring that each citation was correctly linked to the corresponding reference. In addition, we removed the repeatedly cited entries in the related work section to ensure the accuracy and consistency of the citations in the text.

Point 3: Please note the title of the figure. In Analysis 3 of the Definition 4, there are Fig1 and Fig 2, but the introduction part also includes the Fig 1 and Fig 2, which are not the same. In addition, Fig 3 refers to “Urban grid and definition of inflow and outflow”, as well as “The overall framework of the ST-D3DDARN model”, Please correct them.

Response: Thanks to the reviewer for reminding. We apologize for the reading problems caused by the confusion of figure numbers in the paper. We have renumbered the figure titles in the paper so that readers can clearly understand their content.

Point 4: The manuscript controls the parameters and efficiency of the model, where "Time each Epoch" in Table 6 refers to training time? If so, why is the model less time-consuming during training than all comparison models, but more time-consuming during testing? Actually, in practical use of the model, testing time is generally more important.

Response: Thank you for bringing up this important point. "Time each Epoch" refers to the time spent in each Epoch during training. Our model is less time-consuming during training compared to the comparison models, but more time-consuming during testing. This might seem counterintuitive. 

In fact, in order to establish the global spatial dependence of traffic flow, we add a self-attention mechanism to the model, and the parameters of the model are greatly increased due to the existence of multiple fully connected layers in the self-attention mechanism. In our experiments, we found that the large increase in the number of parameters caused by the addition of self-attention mechanism does not lead to a large increase in training time. However, it will cause a significant increase in testing time. We know that model training includes both forward and backward propagation, while model testing only includes forward propagation. The complexity of back propagation calculation is different due to different neural network structure in the model. Therefore, the reason for this phenomenon may be that the large number of parameters generated by the introduction of the self-attention mechanism in our model leads to a large increase in the time of the forward propagation, while its backpropagation complexity is low.

In general, our model has little difference in test time compared with other models, and there is little difference in actual use. In addition, our model requires less computational resources during training and has higher flexibility for practical use.

Point 5: Dividing the transportation network into grids is an approximate method, does it affect prediction accuracy? There are many papers that use graph for traffic prediction, for example “Hierarchical Spatio-Temporal Graph Convolutional Networks and Transformer Network for Traffic Flow Forecasting”, “Knowledge fusion enhanced graph neural network for traffic flow prediction”, you can refer to them.

Response: Thank you for your valuable comments. The references you recommended put forward the HSTGCNT and KE-STGCN models, which show strong effectiveness and robustness in the traffic flow prediction of the road network with irregular topology structure, and their superiority is proved by experiments with the road network traffic flow data collected by sensors and the passenger flow data of subway stations. Dividing the traffic network into grids is a common approximation, and indeed may affect the accuracy of the prediction. However, our work focuses on transforming traffic trajectory data into grid data and then improving the prediction accuracy as much as possible. We mention in the conclusion that future work may involve dividing cities into irregular regions based on their functional characteristics and modeling urban traffic flows using GCN to learn spatio-temporal information better.

In addition, inspired by this comment from you, we realized that we did not compare methods for modeling spatial grid information using GCN. Therefore, we add GCN+LSTM model in our comparison experiments. GCN+LSTM model is a combination model of temporal and spatial. In this method, a graph G = (V, E) is constructed, where V and E represent the set of vertices and edges respectively. Each vertex in a graph G represents a region, and an edge between two vertices indicates that two regions are adjacent. The experimental results are shown in Table 1 and Table 2.

Table 1. Comparison of model prediction results

Methods TaxiBJ BikeNYC

 RMSE MAE RMSE MAE

HA 45.36 22.47 20.33 10.22

ARIMA 22.78 12.72 10.22 5.36

CNN+LSTM 17.29 9.87 5.48 2.83

GCN+LSTM 16.79 9.84 5.28 2.73

ST-ResNet 16.92 9.64 5.27 2.75

DeepSTN+ 16.29 9.27 4.66 2.43

ST-3DNet 16.73 9.69 5.22 2.62

3D-ConvLSTMNet 16.47 9.58 5.14 2.96

MS-ResCnet 15.60 9.15 4.82 2.48

MPCNN 15.77 9.23 5.02 2.47

ST-D3DDARN 14.98 8.91 4.49 2.25

Table 2. Inflow and outflow prediction results in TaxiBJ

Methods Inflow Outflow

 RMSE MAE RMSE MAE

HA 45.27 22.40 45.45 22.54

ARIMA 22.73 12.56 22.82 12.83

CNN+LSTM 17.27 9.85 17.32 9.90

GCN+LSTM 16.75 9.82 16.83 9.86

ST-ResNet 16.89 9.62 16.95 9.66

DeepSTN+ 16.26 9.24 16.32 9.30

ST-3DNet 16.67 9.66 16.78 9.74

3D-ConvLSTMNet 16.39 9.52 16.55 9.64

MS-ResCnet 15.56 9.08 15.64 9.15

MPCNN 15.73 9.19 15.81 9.26

ST-D3DDARN 14.86 8.81 15.11 8.99

Reviewer #2

Overall, in this paper, the authors have made significant contributions to increase the knowledge. 

Response: Thank you for participating in the review of our work.

Point 1: There are a few grammar things that could be improved. Could you double-check the grammar mistakes?

Response: Sorry for the grammar errors. As non-English speakers, the language is truly a problem for us. Thus, according to the reviewer’s suggestion, we first tried our best to check the manuscript carefully, and further, we made revisions by using advanced translation software available in the Chinese market, as well as grammar correction software. Additionally, we sought the assistance of a friend who is proficient in English for further editing. We wish the revised version is more readable than the previous one.

Point 2: Abstract, introduction, contribution and conclusion should be aligned.

Response: Thanks for the constructive suggestion. We have revised inconsistent representations in the paper to ensure consistency between the abstract, introduction, contribution, and conclusion.

---

## [Decision Letter · Decision Letter 1]

30 May 2024

ST-D3DDARN: Urban Traffic Flow Prediction based on Spatio-temporal Decoupled 3D Densenet with Attention ResNet

PONE-D-24-03623R1

Dear Dr. Wei,

We’re pleased to inform you that your manuscript has been judged scientifically suitable for publication and will be formally accepted for publication once it meets all outstanding technical requirements.

Kind regards,

Muhammad Bilal

Academic Editor

PLOS ONE

Reviewers' comments:

Reviewer's Responses to Questions

**Comments to the Author**

1. If the authors have adequately addressed your comments raised in a previous round of review and you feel that this manuscript is now acceptable for publication, you may indicate that here to bypass the “Comments to the Author” section, enter your conflict of interest statement in the “Confidential to Editor” section, and submit your "Accept" recommendation.

Reviewer #2: All comments have been addressed

Reviewer #3: All comments have been addressed

2. Is the manuscript technically sound, and do the data support the conclusions?

Reviewer #2: Yes

Reviewer #3: Yes

3. Has the statistical analysis been performed appropriately and rigorously? 

Reviewer #2: Yes

Reviewer #3: Yes

4. Have the authors made all data underlying the findings in their manuscript fully available?

Reviewer #2: Yes

Reviewer #3: Yes

5. Is the manuscript presented in an intelligible fashion and written in standard English?

Reviewer #2: Yes

Reviewer #3: Yes

6. Review Comments to the Author

Reviewer #2: The authors have already addressed all the comments made at the time of initial review. It is now ready for publication.

Reviewer #3: (No Response)

7. PLOS authors have the option to publish the peer review history of their article (what does this mean?). If published, this will include your full peer review and any attached files.

Reviewer #2: No

Reviewer #3: **Yes: **Dr Muhammad Ramzan

---

## [Editor Report · Acceptance letter]

3 Jun 2024

PONE-D-24-03623R1 

PLOS ONE

Dear Dr. Yang, 

I'm pleased to inform you that your manuscript has been deemed suitable for publication in PLOS ONE. Congratulations! Your manuscript is now being handed over to our production team.

Kind regards, 

on behalf of

Dr. Muhammad Bilal 

Academic Editor

PLOS ONE